# NAVIX: Scaling Minigrid Environments with JAX

**Eduardo Pignatelli**
University College London
e.pignatelli@ucl.ac.uk

**Jarek Liesen**
BCCN Berlin
jarek@bccn-berlin.de

**Robert Tjarko Lange**
Technical University Berlin
robert.t.lange@tu-berlin.de

**Chris Lu**
University of Oxford
christopher.lu@eng.ox.ac.uk

**Pablo Samuel Castro**
Google DeepMind
Mila, Université de Montréal
psc@google.com

**Laura Toni**
University College London
l.toni@ucl.ac.uk

## Abstract

As Deep Reinforcement Learning (Deep RL) research moves towards solving large-scale worlds, efficient environment simulations become crucial for rapid experimentation. However, most existing environments struggle to scale to high throughput, setting back meaningful progress. Interactions are typically computed on the CPU, limiting training speed and throughput, due to slower computation and communication overhead when distributing the task across multiple machines. Ultimately, Deep RL training is CPU-bound, and developing batched, fast, and scalable environments has become a frontier for progress. Among the most used Reinforcement Learning (RL) environments, Minigrid is at the foundation of several studies on exploration, curriculum learning, representation learning, diversity, meta-learning, credit assignment, and language-conditioned RL, and still suffers from the limitations described above. In this work, we introduce NAVIX[1], a re-implementation of Minigrid in JAX. NAVIX achieves over $160\,000\times$ speed improvements in batch mode, supporting up to 2048 agents in parallel on a single Nvidia A100 80 GB. This reduces experiment times from one week to 15 minutes, promoting faster design iterations and more scalable RL model development.



|     |     |     |     |     |
| --- | --- | --- | --- | --- |
| (a) 128.98× | (b) 26.47× | (c) 41.17× | (d) 19.72× | (e) 45.71× |

Figure 1: Speedups for five of the NAVIX environments with respect to their Minigrid equivalent, using the protocol in Section 4.1. (a) `Empty-8x8-v0`, (b) `DoorKey-8x8-v0`, (c) `Dynamic-Obstacles-8x8-v0`, (d) `KeyCorridorS3R3-v0`, (e) `LavaGapS7-v0`.

---

[1]https://github.com/epignatelli/navix

39th Conference on Neural Information Processing Systems (NeurIPS 2025) Track on Datasets and Benchmarks.

# 1 Introduction

Deep RL is notoriously sample inefficient (Kaiser et al., 2019; Wang et al., 2021; Johnson et al., 2016; Küttler et al., 2020). Depending on the complexity of the environment dynamics, the observation space, and the action space, agents often require between $10^7$ to $10^9$ interactions or even more for training up to a good enough policy. Therefore, as Deep RL moves towards tackling more complex environments, leveraging efficient environment implementations is an essential ingredient of rapid experimentation and fast design iterations.

However, while the efficiency and scalability of solutions for *agents* have improved massively in recent years (Schulman et al., 2017; Espeholt et al., 2018; Kapturowski et al., 2018), especially due to the scalability of the current deep learning frameworks (Abadi et al., 2016; Paszke et al., 2019; Ansel et al., 2024; Bradbury et al., 2018; Sabne, 2020), environments have not kept pace. They are mostly based on CPU, cannot adapt to different types of devices, and scaling often requires complex distributed systems, introducing design complexity and communication overhead. Overall, deep RL experiments are CPU-bound, limiting both speed and throughput of RL training.

Recently, a set of GPU-based environments (Freeman et al., 2021; Lange, 2022; Weng et al., 2022; Koyamada et al., 2023; Rutherford et al., 2023a; Nikulin et al., 2023; Matthews et al., 2024; Bonnet et al., 2024; Lu et al., 2023; Liesen et al., 2024b) and frameworks (Hessel et al., 2021; Lu et al., 2022; Liesen et al., 2024a; Toledo, 2024; Nishimori, 2024; Jiang et al., 2023) has sparked raising interest, proposing JAX-based, batched implementations of common RL environments that can significantly increase the speed and throughput of canonical Deep RL algorithms. This enables large-scale parallelism, allowing the training of thousands of agents in parallel on a single accelerator, significantly outperforming traditional CPU-based environments, and fostering meta-RL applications.

In this work, we build on this trend and focus on the Minigrid suite of environments (Chevalier-Boisvert et al., 2024), due to its central role in the Deep RL literature. MiniGrid is fundamental to many studies. For instance, Zhang et al. (2020); Zha et al. (2021); Mavor-Parker et al. (2022) used it to test new exploration strategies; Jiang et al. (2021) for curriculum learning; Zhao et al. (2021) for planning; Paischer et al. (2022) for representation learning, Flet-Berliac et al. (2021); Guan et al. (2022) for diversity. Parisi et al. (2021) employed Minigrid to design meta and transfer learning strategies, and Mu et al. (2022) to study language grounding.

However, despite its ubiquity in the Deep RL literature, Minigrid faces the limitations of CPU-bound environments. We bridge this gap and propose NAVIX, a reimplementation of Minigrid in JAX that leverages JAX's intermediate language representation to migrate the computation to different accelerators, such as GPUs, and TPUs.

Our results show that NAVIX is over $44\times$ times faster than the original Minigrid implementation, in common Deep RL settings (Section 4.1), $5.4$ times faster when running a single environment (Appendix I), and increases the throughput by over $10^6\times$ (Section 4.2). Composed together, they produce speed-ups of over $160\,000\times$ (Section 4.2), turning 1-week experiments into 15 minutes ones. We show the scaling ability of NAVIX by training over 2048 PPO agents in parallel (Section 4.2), each using their own subset of environments, all on a single Nvidia A100 80 GB.

The main contributions of this work are the following:

1. A fully JAX-based implementation of environment configurations that reproduces exactly the original Minigrid Markov Decision Processes (MDPs) and Partially-observable MDPs (POMDPs).

2. A description of the design philosophy, the design pattern and principles, the organisation, and the components of NAVIX, which, together with the online documentation, form an instruction manual to use and extend NAVIX.

3. A set of RL algorithm baselines for all environments in Section 4.3.

# 2 Related work

**JAX-based environments.** The number of JAX-based reimplementations of common environments is in a bullish trend. Freeman et al. (2021) provide a fully differentiable physics engine for robotics, including MJX, a reimplementation of MujoCo (Todorov et al., 2012). Lange (2022) reimplements

several gym (Brockman et al., 2016) environments, including classic control, Bsuite (Osband et al., 2020), and MinAtar (Young & Tian, 2019),

Koyamada et al. (2023) reimplement many board games, including backgammon, chess, shogi, and go. Lu et al. (2023) provides JAX implementations of POPGym (Morad et al., 2023), which contains partially-observed RL environments. Matthews et al. (2024) reimplement Crafter (Hafner, 2021). Bonnet et al. (2024) provides JAX implementations of combinatorial problems frequently encountered in industry, including bin packing, capacitated vehicle routing problem, PacMan, Sokoban, Snake, 2048, Sudoku, and many others. Rutherford et al. (2023b) reimplement a set of multi-agent environments, including a MiniGrid-inspired implementation of the Overcooked benchmark.

Yet, none of these works proposes a reimplementation of Minigrid. Weng et al. (2022) is the only one providing a single environment of the suite, *Empty*, but it is only one of the many, most commonly used environments of the suite, and arguably the simplest one.

**Batched MiniGrid−like environments.** Two works stand out for they aim to partially reimplement MiniGrid. Jiang et al. (2023) present AMaze, a fully batched implementation of a partially observable maze environment, with MiniGrid−like sprites and observations. However, like Weng et al. (2022), the work does not reimplement the full Minigrid suite. Nikulin et al. (2023) proposes XLand-Minigrid, a suite of grid-world environments for meta RL. Like (Jiang et al., 2023), XLand-Minigrid reproduces Minigrid-like observations but focuses on designing a set of composable rules that can be used to generate a wide range of environments, rather than mirroring the original Minigrid suite while reimplementing it in JAX.

To conclude, Minigrid is a fundamental tool for Deep RL experiments, at the base of a high number of studies, as we highlighted in Section 1. It is easy to use, easy to extend, and provides a wide range of environments of scalable complexity that are easy to inspect for a clearer understanding of an algorithm dynamics, pitfalls, and strengths.

Nevertheless, none of the works above provides a full, batched reimplementation of Minigrid in JAX that mirrors the original suite in terms of environments, observations, state transitions, and rewards. Instead, we propose a full JAX-based reimplementation of the Minigrid suite with identical semantics for observations, actions, rewards, and terminations.

## 3 NAVIX: design philosophy and principles

In this section we describe:

- *(i)* the design philosophy and pattern of NAVIX in Section 3.1, and
- *(ii)* the design principles at its foundation in Sections 3.2.1 and 3.2.2.

In particular, in Section 3.2.2, we highlight *why* a JAX port of Minigrid is not trivial. Among others, the obstacles to transform a stateful program, where a function is allowed to change elements that are not an input of the function, into a stateless one, where the outputs of functions depend solely on the inputs; and the restrictions in the use of `for` loops and control flow primitives, such as `if` statements.[2]

### 3.1 Design pattern

NAVIX is broadly inspired by the ECSM, a design pattern widely used in video game development. In an ECSM, entities – the *objects* on the grid in our case – are composed of components – the *properties* of the object. Each property holds data about the entity, which can then be used to process the game state. For example, an entity `Player` is composed of components `Positionable`, `Holder`, `Directional`, each of which injects properties into the entity: the `Positionable` component injects the `Position` property, holding the coordinates of the entity (e.g., a player, a door, a key) on the grid, the `Holder` component injects the `Pocket` property, holding the id of the entity that the agent holds, and so on. A full list of components and their properties is provided in Table 1. This compositional layout allows to easily generate the wide range of combinations of tasks that Minigrid offers, and to easily extend the suite with new environments.

---

[2]See `https://jax.readthedocs.io/en/latest/notebooks/Common_Gotchas_in_JAX.html`.

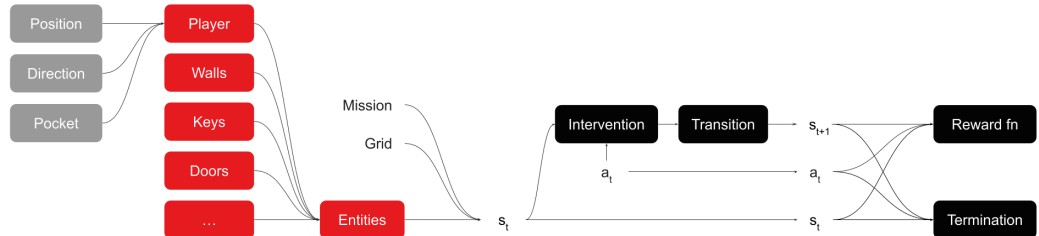

Figure 2: Information flow of the Entity-Component-System Model (ECSM) in NAVIX. Entities (*Player, Walls, Keys, Doors, ...*) are composed of components (*Position, Direction, Pocket*), which hold the data of the entity. Systems (*Intervention, Transition, Rewards, Terminations*) are functions that operate on the collective state of all entities and components.

| Component | Property | Shape | Description |
|---|---|---|---|
| Positionable | Position | f32[2] | *Coordinates of the entity on the grid.* |
| Directional | Direction | i32[] | *Direction of the entity.* |
| HasColour | Colour | u8[] | *Colour of the entity.* |
| Stochastic | Probability | f32[] | *Probability that the entity emits an event.* |
| Openable | State | bool[] | *State of the entity, e.g., open or closed.* |
| Pickable | Id | i32[] | *Id of the entity that the agent can pick up.* |
| HasTag | Tag | i32[] | *Categorical value identifying the entity class.* |
| HasSprite | Sprite | u8[32x32x3] | *Sprite of the entity in RGB format.* |
| Holder | Pocket | i32[] | *Id of the entity that the agent holds.* |

Table 1: List of **Components** in NAVIX. Each component provides a property (or a set of). These properties hold the data that can be accessed and manipulated by the systems (see Table 3) to provide observations, rewards, and state transitions.

| Entity | Components | Description |
|---|---|---|
| Wall | [HasColour] | *An entity that blocks the agent's movement.* |
| Player | [Directional, Holder] | *An entity that can interact with the environment.* |
| Goal | [HasColour, Stochastic] | *An entity that the agent can to reach to receive a reward.* |
| Key | [Pickable, HasColour] | *An entity that can be picked up. Can open doors.* |
| Door | [Openable, HasColour] | *An entity that can be opened and closed by the agent.* |
| Lava | [] | *An entity that the agent has to avoid.* |
| Ball | [HasColour, Stochastic] | *An entity that the agent can push.* |
| Box | [HasColour, Holder] | *An entity that the agent can push.* |

Table 2: List of **Entities** in NAVIX, together with the components that characterise them. By default, all entities already possess Positionable, HasTah, and HasSprite components, in addition to those reported in the table.

| System | Function | Description |
|---|---|---|
| Intervention | $I : \mathcal{S} \times \mathcal{A} \to \mathcal{S}$ | *Updates the state according to the agent's actions.* |
| Transition | $P : \mathcal{S} \times \mathcal{A} \to \mathcal{S}$ | *Updates the state according to the MDP dynamics.* |
| Observation | $O : \mathcal{S} \to \mathcal{O}$ | *The observation kernel;* |
| Reward | $R : \mathcal{S} \times \mathcal{A} \times \mathcal{S} \to \mathbb{R}$ | *The Markovian reward function.* |
| Termination | $\gamma : \mathcal{S} \times \mathcal{A} \times \mathcal{S} \to \mathbb{B}$ | *The termination function.* |

Table 3: List of **Systems** in NAVIX. A state $s \in \mathcal{S}$ is a tuple containing: the set of entities, the static grid layout, and the mission of the agent.

Entities are then processed by *systems*, which are functions that operate on the collective state of all entities and components. For example, the *decision* system may update the state of the entities according to the actions taken by a player. The *transition* system may update the state of the entities according to the MDP state transitions. The *observation* system generates the observations that the agents receive, and the *reward* system computes the rewards that the agents receive, and so on. We provide a full list of implemented systems in Appendix A.

To develop a better intuition of what these elements are and how they interact, Figure 2 shows the information flow of the ECSM in NAVIX.

## 3.2 Design principles

On this background, two principles are at the foundation of NAVIX, and the key aspects that characterise it:

*(i)* NAVIX aims to exactly match the semantics of Minigrid (Section 3.2.1) with identical observations, actions, rewards, and terminations;

*(ii)* every environment is fully jittabile, i.e., that can be compiled into more efficient instructions using JAX, and differentiable (Section 3.2.2), to exploit the full set of features that JAX offers.

### 3.2.1 NAVIX matches MiniGrid

NAVIX matches the original Minigrid suite in terms of environments, observations, state transitions, rewards, and actions. We include the most commonly used environments of the suite (see Table 14, Appendix I), and provide a set of baselines for the implemented environments in see Section 4 and Table 14, Appendix I.

Formally, a NAVIX environment is a tuple $\mathcal{M} = (h, w, T, \mathcal{O}, \mathcal{A}, \mathcal{R}, d, O, R, \gamma, P)$. Here, $h$ and $w$ are the height and width of the grid, $T$ is the number of timesteps before timeout; $\mathcal{O}$ is the observation space, $\mathcal{A}$ is the action space, $\mathcal{R}$ is the reward space; $\gamma$ is the discount factor. $O$ is the observation function, $R$ is the reward function, $d$ is the termination function, and $P$ is the transition function.

**Reward functions** A key difference between NAVIX and MiniGrid, by design, is that the latter uses a non-Markovian reward function. In fact, Minigrid dispenses a reward of $0$ everywhere, except at task completion, where it is inversely proportional to the number of steps taken by the agent to reach the goal:

$$r_t = R(s_t, a, s_{t+1}) - 0.9 * (t + 1)/T, \tag{1}$$

Here $R$ is the reward function, $s_t$ is the state at time $t$, $a$ is the action taken at time $t$, $s_{t+1}$ is the state at time $t + 1$, and $T$ is the number of timesteps before timeout. Notice the dependency on the number of steps $t$, which makes the reward non-Markovian.

The use of a non-Markovian reward function is not a mild assumption as most of the RL algorithms assume Markov rewards (Schulman et al., 2017; Haarnoja et al., 2018b; van Hasselt et al., 2016). This might call into question the validity of the historical results obtained with MiniGrid, and the generalisation of the results to other environments.

However, the necessity to align the Minigrid reward function with Markov assumptions is in stark contrast with the principle to reproduce the exact reward semantics of MiniGrid. Since this is a point of difference that might invalidate our claim that NAVIX is a semantically compatible replacement for MiniGrid, we leverage the modularity of NAVIX and supply two ready-to-use reward functions. These functions are variables that can be easily changed at the time of the creation of the environment. In the first function, we depart from the original Minigrid reward function and use a Markovian reward function, which is $0$ everywhere, $1$ at task completion, and $-c$ every at every timestep if the agent performs an action different from the `do-nothing` action. Here, where $c$ is a constant action cost. In the second version, we replicate the reward function of Minigrid in Equation (1). We analyse the impact of Markovianity on the training of a PPO agent in Appendix B by measuring how the task completion rate varies during training for each reward function.

### 3.2.2 Stateless and fully jittable

While we aim to match Minigrid in terms of environments, observations, state transitions, rewards, and actions, the API of NAVIX is different, as it must align with JAX requirements for the environment to be fully jittable. In fact, NAVIX environments can be compiled using Accelerated Linear Algebra (XLA) – an open-source compiler for machine learning that optimises Python code for high-performance execution across different hardware platforms including GPUs, CPUs, TPUs and other ML accelerators. This includes both simply jitting – i.e., compiling just-in-time – the `step` function, and jitting the entire training sequence (Lu et al., 2022), assuming that the agent is also implemented in JAX. XLA compilation increases the throughput of experiments massively, allowing for the training of thousands of agents in parallel on a single accelerator, compared to a few that are possible with traditional CPU-based environments. We show the scalability of NAVIX in Section 4.

For environments to be fully jittable, the computation must be stateless. For this reason, we need to define an environment *state-object*: the *timestep*. The timestep is a tuple $(t, o_t, a_t, r_{t+1}, \gamma_{t+1}, s_t, i_{t+1})$, where $t$ is the current time – the number of steps elapsed from the last reset – $o_t$ is the observation at time $t$, $a_t$ is the action taken after $o_t$, $r_{t+1}$ is the reward received after $a_t$, $\gamma_{t+1}$ is the termination signal after $a_t$, $s_t$ is the true state of the environment at time $t$, and $i_{t+1}$ is the info dictionary, useful to store accumulations, such as returns.

This structure is necessary to guarantee the same return schema for both the `step` and the `reset` methods, and allows the environment to autoreset, and avoid conditional statements in the agent code, which would prevent the environment from being fully jittable.

At the beginning of the episode, the agent samples a starting state from the starting distribution $P_0 : \mathcal{S} \rightarrow \mathcal{S}$ using the `reset(key)` method, where `key` is a key for a stateless random number generator. Since there is no action and reward at the beginning of the episode, we pad with $-1$ and $0$, respectively. Given an action $a_t$, the agent can interact with the environment by calling the `step(timestep, action, key)` method. The agent then receives a new state of the environment (a new timestep) and can continue to interact as needed. Code 1 shows an example of how to interact with a jitted NAVIX environment. More examples will be provided online.

```python
import navix as nx

# init a NAVIX environment
env = nx.make("Navix-KeyCorridorS6R3-v0")

# sample a starting state
timestep = env.reset(key)
for _ in range(1000):
    # sample a random key
    key, subkey = jax.random.split(key)
    # sample a random action
    action = jax.random.randint(subkey, (1,), 0, 4)
    # interact with the environment
    timestep = jax.jit(env.step)(timestep, action)  # autoresets when done
```

Code 1: Example code to interact with a jitted NAVIX environment.

Notice that the syntax is similar to the original MiniGrid, including the environment *id*, which simply replaces "MiniGrid" with "Navix". The only differences are in the use of an explicit random key for the stateless random number generator, and the fact that the `step` method also takes the current timestep as input, to guarantee the statelessness of the computation.

The schema in Code 1 is an effective template for any kind of agent implementation, including non JAX-jittable agents. However, while this already improves the speed of environment interactions compared to MiniGrid, as shown in Section 4.1, the real speed-up comes jitting the whole iteration loop. In Appendix F we provide additional reusable patterns that are useful in daily RL research, such as how to jit the training loop, how to parallelise the training of multiple agents, and how to run hyperparameter search in batch mode.

In addition, in Appendix H we provide a guide on how to extend NAVIX, including new environments, new observations, new rewards, and new termination functions. This is a fundamental aspect to reflect the flexibility of the original Minigrid suite, which is easy to extend and modify.

## 4   Experiments

This section aims to show the advantages of NAVIX compared to the original Minigrid implementation, and provides the community with a set of baselines for all environments. It does the former by comparing the two suites, for all environments, both in terms of speed improvements and throughput. For the latter, we train a set of baselines for all environments, and provide a scoreboard that stores the results for all environments. All experiments are run on a single Nvidia A100 80Gb, and Intel(R) Xeon(R) Silver 4310 CPU @ 2.10GHz and 128Gb of RAM.

### 4.1   Speed

We first benchmark the raw speed improvements of NAVIX compared to the original Minigrid implementation, in the most common settings. For each NAVIX environment and its Minigrid equivalent, we run $1K$ steps with 8 parallel environments, and measure the wall time of both. Notice that this is the mere speed of the environment, and does not include the agent training.

We show results in Figure 3, and observe that NAVIX is over $44\times$ times faster than the original Minigrid implementation on average. These improvements are due to both the migration of the computation to the GPU via XLA, which optimises the computation graph for the specific accelerator, and the batching of the environments. In Figure 15, Appendix I we ablate the batching, with no parallel environments, and show that the biggest contribution for the speedup is due to efficient batching.

To better understand how the speedup varies with the number of training steps, and to make sure that the $1K$ steps used in the previous experiment are representative of the general trend, we measure the speed improvements for different lengths of the training runs. We run $1K$, $10K$, $100K$, and $1M$ steps for the `MiniGrid-Empty-8x8-v0` environment and its NAVIX equivalent, and measure the wall time of both.

Results in Figure 4 show that NAVIX is consistently faster than the original Minigrid implementation, regardless of the number of steps. Both Minigrid and NAVIX show a linear increase in the wall time with the number of steps.

### 4.2   Throughput

While NAVIX provides speed improvements compared to the original Minigrid implementation, the real advantage comes from the ability to perform highly parallel training runs on a single accelerator. In this experiment, we test how the computation scales with the number of environments.

We first test the limits of NAVIX by measuring the computation while varying the number of environments that run in parallel. MiniGrid uses `gymnasium`, which parallelises the computation with *Python*'s multiprocessing library. NAVIX, instead, uses JAX's native `vmap`, which directly vectorises the computation. We confront the results with the original Minigrid implementation, using the `MiniGrid-Empty-8x8-v0` environment.

Results in Figure 5 show that the original Minigrid implementation cannot scale beyond 16 environments on 128GB of RAM, for which it takes around $1s$ to complete $1K$ unrolls. On the contrary, NAVIX can run up to $2^{21}$ (over $2M$) environments in parallel on the same hardware, with a wall time almost always below $1s$. In short, NAVIX achieves a throughput over $10^5$ orders of magnitude higher than the original Minigrid implementation.

Secondly, we simulate the very common operation of training many PPO agents, each with their own subset of 16 environments. However, with NAVIX, we can do this in parallel. We use the `Empty-8x8-v0` environment, and train the agent for $1M$ steps. Results are shown in Figure 6.

Overall, we observe that, on a single NVIDIA A100 80GB, stepping a batch of $2048 \times 16 = 32\,768$ NAVIX environments for $10^6$ transitions takes $49s$. This is a throughput of $2048 \times 16 \times 1M/49s = 6.7 \times 10^8$ environment steps/s. On the other hand, the original Minigrid reference with a CleanRL

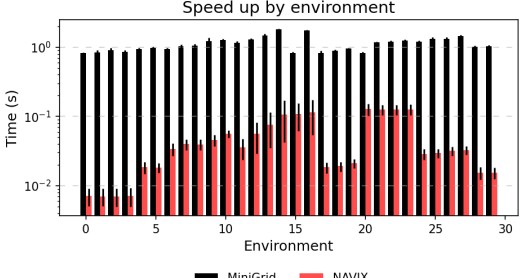

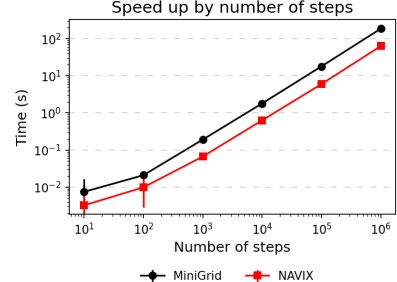

Figure 3: Speedup of NAVIX compared to the original Minigrid implementation, for the implemented environments. The identifiers on the x-axis correspond to the environments as reported in Table 13, Appendix I. Results are the average across 5 runs.

Figure 4: Variation of the speedup of NAVIX compared to the original Minigrid implementation.

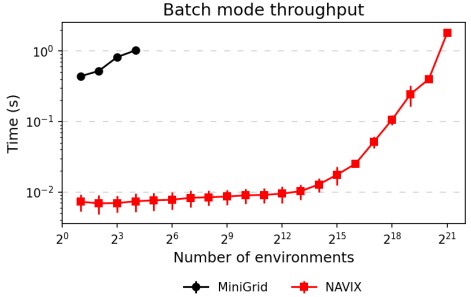

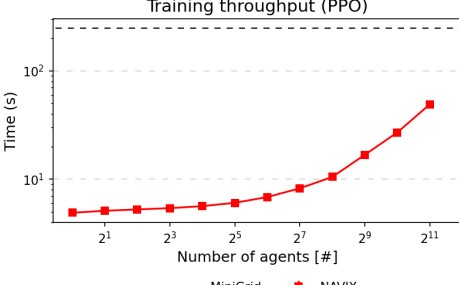

Figure 5: Wall time of $1K$ unrolls for both NAVIX and Minigrid in batch mode.

Figure 6: Computation costs with growing batch sizes.

baseline, executed on the same host with a dual-socket Intel® Xeon® Silver 4310 (24 cores, 48 threads at 2.10 GHz), takes $240s$ for a *single* PPO agent. This is a throughput of $1 \times 1M/240s = 4.2 \times 10^3$ environment steps/s. In other words, NAVIX achieves a speedup of $6.7 \times 10^8/4.2 \times 10^4 = \approx 160\,000\times$. All timings exclude the initial XLA compilation and a one-episode warm-up for fairness. This comparison fixes the compute budget to a single node to highlight the throughput boost that NAVIX unlocks. Matching the same wall-clock throughput with CPU-only Minigrid would require on the order of $10^5$ concurrently active environments spread across many machines – incurring in network synchronisation overheads that our single-GPU setup avoids, and costs are that far beyond those of a single NVIDIA A100 80 GB node.

To complement these results, and give readers a more comprehensive expectation of NAVIX's performance, we also performed the speed and throughput experiments on consumer-grade hardware. Results in Figures 9, 10, 11, and 12, Appendix C, confirm this trends of this section.

## 4.3 Baselines

We provide additional baselines using the implementations of PPO (Schulman et al., 2017), Double DQN (DDQN) (Hasselt et al., 2016), Soft Actor Critic (SAC) (Haarnoja et al., 2018a), IQN (Dabney et al., 2018), and PQN (Gallici et al., 2025) in Rejax (Liesen et al., 2024a)[3]. We optimize hyperparameters (HP) for each algorithm and environment combination using 32 iterations of random search. Each HP configuration is evaluated with 16 different initial seeds. The HP configuration with the highest average final return is selected. The specific hyperparameters we searched for are shown in Table 12, Appendix E. The definitive list of hyperparameters used in the experiments can be found in Section E.

---

[3]https://github.com/keraJLi/rejax

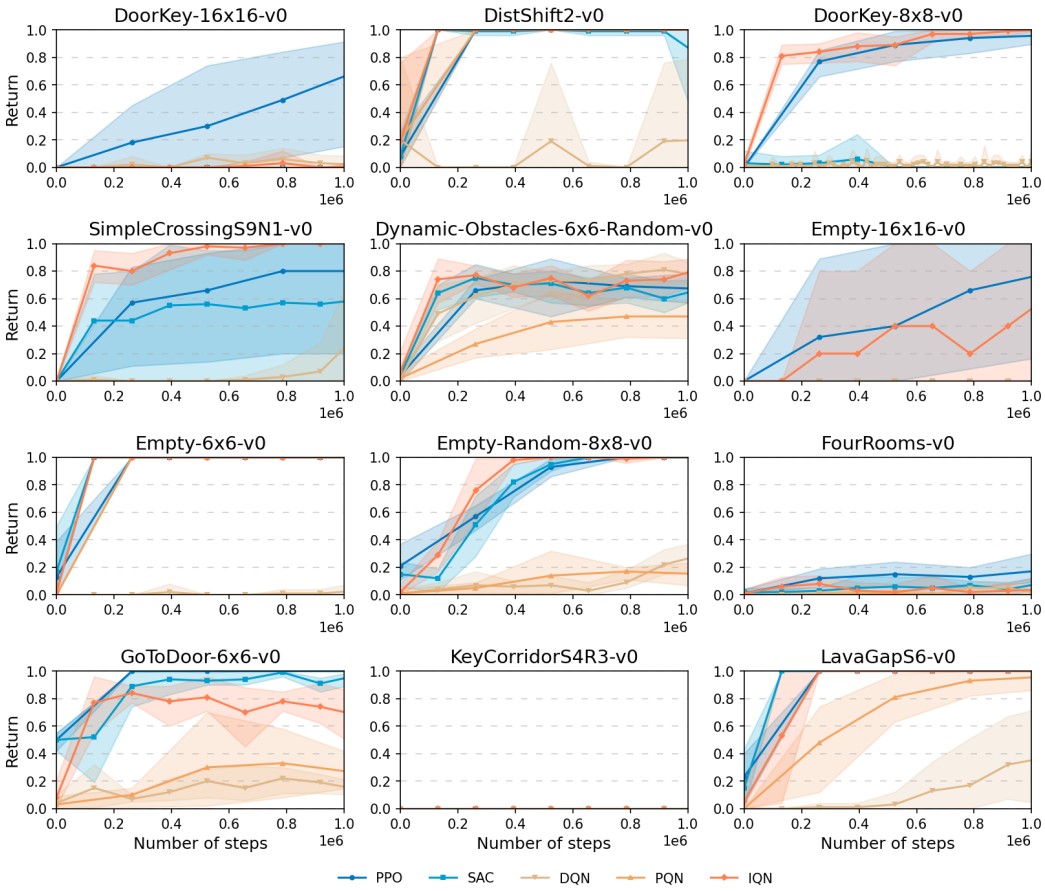

Figure 7: Episodic returns for a sample of NAVIX environments for DDQN, PPO and SAC baselines. Lines are average over 32 seeds, and shaded areas show 5-95 percentile confidence intervals.

We run the baselines for $1M$ steps, across 32 seeds, with the tuned hyperparameters for the environments shown in Figure 7. All algorithms use networks with two hidden layers of 64 units. Instead of alternating between a single environment step and network update, DQN and SAC instead perform 128 parallel environment steps and 128 network updates, each with a new minibatch. We found that this significantly improves the runtime while leaving the final performance unaffected.

## 5 Broader Impact

NAVIX lowers the hardware barrier to entry for RL research by providing a GPU-batched, JAX-native re-implementation of the Minigrid environment suite. Because NAVIX remains fully open-source (Apache 2.0) and can be run on commodity GPU hardware, instructors and students at resource-constrained institutions can reproduce state-of-the-art Minigrid agents on a single desktop GPU, democratizing access to RL education and experimentation.

However, higher-throughput simulation can also potentially accelerate the development cycle of autonomous navigation agents that could be deployed for disallowed or malicious purposes. Nevertheless, NAVIX itself does not contain any trained policies, faster training may facilitate downstream misuse.

## 6 Conclusions

We introduced NAVIX, a reimplementation of the Minigrid environment suite in JAX that leverages JAX's intermediate language representation to migrate the computation to different accelerators, such

as GPUs and TPUs. We described the design pattern, highlighting the connections to the ECSM, and the correspondence between the structure of its functions and the mathematical formalism of RL. We presented the environment interface, the list of available environments.

We showed the speed improvements of NAVIX compared to the original Minigrid implementation, and the scalability of NAVIX with respect to the number of agents that can be trained in parallel, or the number of environments that can be run in parallel.

Overall, NAVIX can be over $160\,000\times$ faster than the original Minigrid implementation, turning 1-week experiments into 15-minute ones. We noticed that these results assume access to a modern GPU, and extending NAVIX to the few remaining Minigrid variants and to training pipelines that cannot JIT-compile the full loop is left for future work.

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

# A  Details on NAVIX systems

Systems are *functions* that operate on the collective state of all entities, defining the rules of the interactions between them. In designing NAVIX, we aimed to maintain a bijective relationship between the systems and their respective mathematical formalism in RL. This makes it easier to translate the mathematical formalism into code, and vice versa, connecting the implementation to the theory. NAVIX includes the following systems:

1. `Intervention`: a function that updates the state of the entities according to the actions taken by the agents.
2. `Transition`: a function that updates the state of the entities according to the MDP state transitions.
3. `Observation`: a function that generates the observations that the agents receive.
4. `Reward`: a function that computes the rewards that the agents receive.
5. `Termination`: a function that determines if the episode is terminated.

We now describe the systems formally.

The intervention is a function $I : S \times A \rightarrow S$ that updates the state of the entities according to the actions taken by the agents. This corresponds to the canonical decision in an MDP.

The transition is a function $\mu : S \times A \rightarrow S$ that updates the state of the entities according to the MDP state transitions. This corresponds to the canonical state transition kernel in an MDP.

The observation is a function $O : S \rightarrow O$ that generates the observations that the agents receive. NAVIX includes multiple observation functions, each generating a different type of observation, for example, a first-person view, a top-down view, or a third-person view, both in symbolic and pixel format. We provide both full and partial observations, allowing to cast a NAVIX environment both as an MDP or as a POMDP, depending on the needs of the algorithm. This follows the design of the original MiniGrid suite.

The reward is a function $R : S \times A \rightarrow \mathbb{R}$ that computes the rewards that the agents receive. Likewise, the termination is a function $\gamma : S \rightarrow \{0, 1\}$ that determines if the episode is terminated. We include both the reward and the termination functions necessary to reproduce all MiniGrid environments. Both these systems rely on the concept of *events*, representing a goal to achieve. An *event* is itself an entity signalling that a particular state of the environment has been reached. For example, it can indicate that the agent has reached a particular cell, has picked up a particular object, or that the agent performed a certain action in a particular state.

We provide a summary of the implemented systems in NAVIX in Tables 4, 5, and 6 for the observation, reward, and termination systems, respectively.

| Observation function | Shape | Description |
|---|---|---|
| `symbolic` | `i32[H, W, 3]` | *The canonical grid encoding observation from MiniGrid.* |
| `symbolic_first_person` | `i32[R, R, 3]` | *A first-person view of the environment in symbolic format.* |
| `rgb` | `u8[32 * H, 32 * W, 3]` | *A fully visible image of the environment in RGB format.* |
| `rgb_first_person` | `u8[32 * R, 32 * R, 3]` | *A first-person view of the environment in RGB format.* |
| `categorical` | `i32[H, W]` | *A grid of entities ID in the environment.* |
| `categorical_first_person` | `i32[R, R]` | *A first-person view of the grid of entities ID.* |

Table 4: Implemented observation functions in NAVIX.

| Reward function | Description |
|---|---|
| `on_goal_reached` | $+1$ *when a Goal entity and a Player entity have the same position* |
| `on_lava_fall` | $-1$ *when a Lava entity and a Player entoty have the same position* |
| `on_door_done` | $+1$ *when the* `done` *action is performed in front of a door with the colour specific in the* `mission` |
| `free` | 0 *everywhere* |
| `action_cost` | $-cost_a$ *at every action taken, except* `done` |
| `time_cost` | $-cost_t$ *at every step* |

Table 5: Implemented reward functions in NAVIX.

| Termination function | Description |
|---|---|
| `on_goal_reached` | *Terminates when a Goal entity and a Player entity have the same position* |
| `on_lava_fall` | *Terminates when a Lava entity and a Player entity have the same position* |
| `on_door_done` | *Terminates when the* `done` *action is performed in front of a door with the colour specific in the* `mission` |
| `free` | 0 *everywhere* |

Table 6: Implemented termination functions in NAVIX.

# B   Impact of reward Markovianity on PPO training

As described in Section 3.2.1, NAVIX replicates the semantics of MiniGrid in terms of environments, observations, state transitions rewards, and actions. However, while developing, we noticed that MiniGrid's reward function is non-Markovian, despite most of the RL algorithms assuming Markov rewards (Schulman et al., 2017; Haarnoja et al., 2018b; van Hasselt et al., 2016). This might call into question the validity of the historical results obtained with MiniGrid, and the generalisation of the results to other environments.

Here, we analyse the impacts of Markovianity on the training of a PPO agent, and how the task completion rate varies during training, with a Markov and a non-Markov reward function. We use the following reward function, respectively:

$$r_t = R(s_t, a, s_{t+1}) \tag{2}$$

$$r_t = R(s_t, a, s_{t+1}) - 0.9 * (t+1)/T. \tag{3}$$

Equation (2) represents the Markovian reward function, while Equation 3 the non-Markovian one.

Results in figure 8 shows that, for some environments, in particular those where PPO converges to near-optimality, the success rate presents a higher variance with non-Markov rewards. To avoid

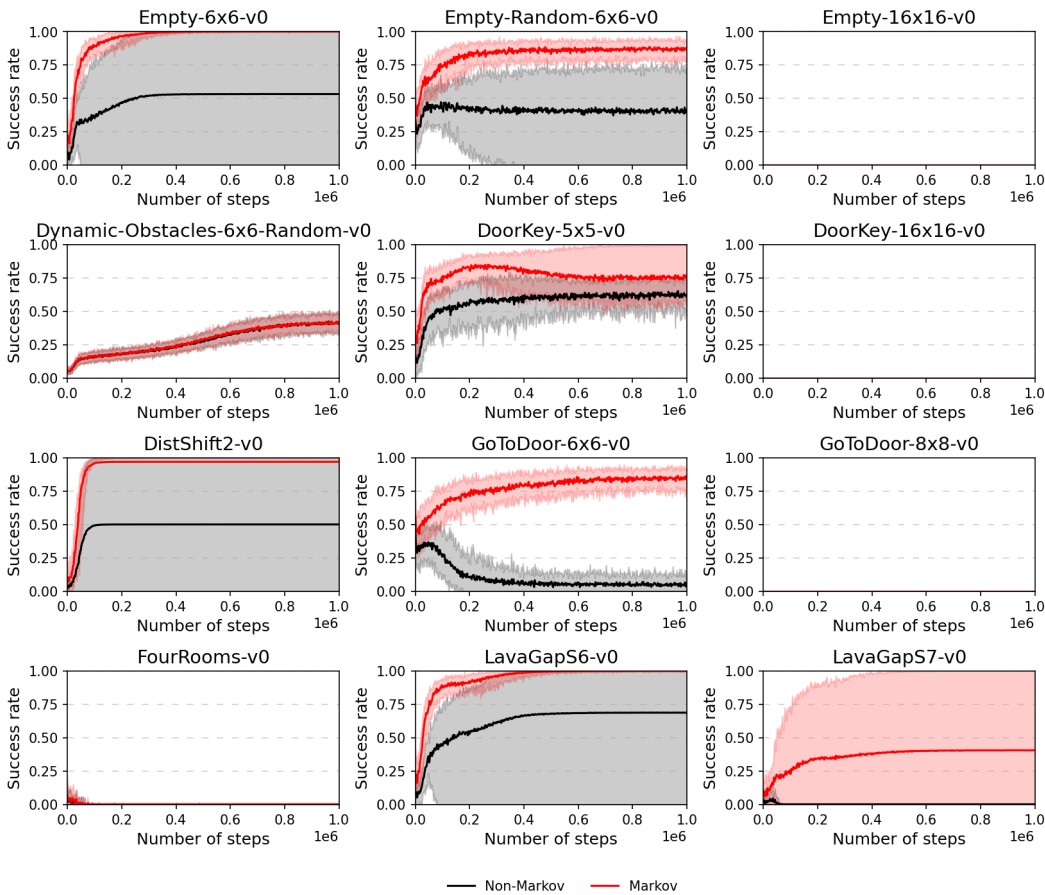

Figure 8: Task completion rate of a PPO agent across environments with a Markov (red) and a non-Markov (black) reward function.

failures when reproducing prior MiniGrid studies, this experiment calls researchers to pay particular attention to choose the original non-Markovian reward function of MiniGrid, also available in NAVIX.

## C  Performance on consumer-grade hardware

Despite NAVIX being designed for training at scale, to give readers a more comprehensive expectation of its performance, we tested the speed and throughput gain in Section 4.1 and 4.2 on a consumer-grade level hardware. The experiment is carried on a consumer desktop with an i7-11700 @ 2.50GHz CPU, 32Gb of RAM, and an Nvidia RTX A4000 with 16Gb of VRAM. All results are averaged across 5 runs.

The evidence in Figure 9 and 10 confirms the trend of Section 4.1. NAVIX is over 7 times faster than the original MiniGrid implementation on average across environments.

Figure 11 and 12, instead, replicate the experiments in Section 4.2. Results show consistent performance on a consumer-grade machine. On a consumer-grade machine, NAVIX can run up to $2^{1}8$ (over $260K$) environments in parallel on a single Nvidia RTX A4000 with 16Gb of VRAM, and over $512$ full PPO agents.

While it is expected that consumer-hardware cannot keep up with the high-end hardware designed for computation at scale, NAVIX still shows a speedup of approximately $34\,000\times$.

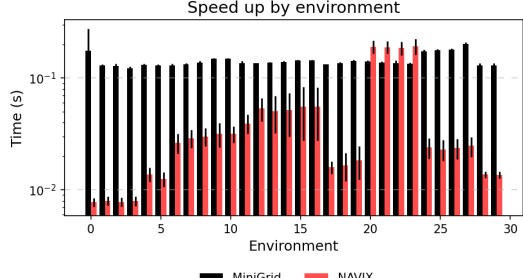

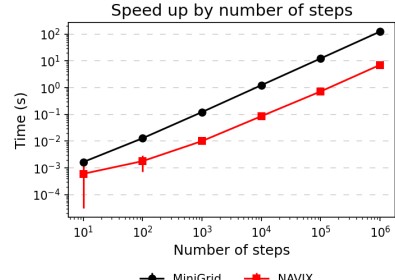

Figure 9: Speedup of NAVIX compared to the original Minigrid implementation on a commercial-grade desktop. The identifiers on the x-axis correspond to the environments as reported in Table 13, Appendix I. Results are the average across 5 runs.

Figure 10: Variation of the speedup of NAVIX compared to the original Minigrid implementation on a consumer-grade machine.

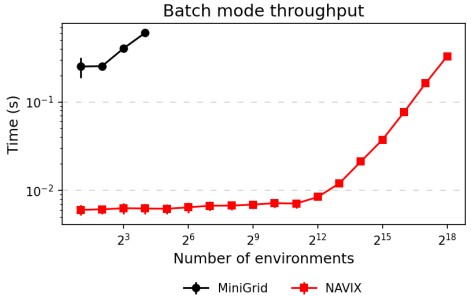

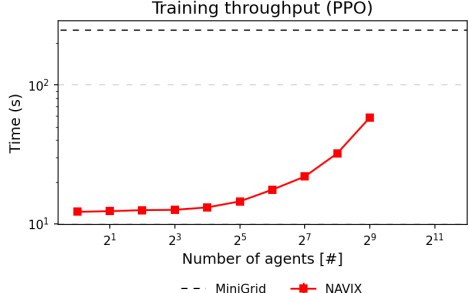

Figure 11: Wall time of $1K$ unrolls for both NAVIX and MiniGrid in batch mode on a consumer-grade machine.

Figure 12: Computation costs with growing batch sizes on a consumer-grade machine. The horizontal dashed line shows the MiniGrid time to train a single PPO agent.

## D  Impact of the design pattern

The speedup of NAVIX does not only depend on JAX, but also on the transition to an ECS design pattern. To marginalise the performance gains over the design patter, we have compared the performance of NAVIX on CPU and GPU: the exact same Python program, with the exact same design pattern, but compiled and run on different accelerators.

We measure speed and throughput as described in Section 4.1 and Section 4.2. The experiment is carried on a desktop with an i7-11700 @ 2.50GHz CPU, 32Gb of RAM, and an Nvidia RTX A4000 with 16Gb of VRAM. Figures 13 and 14 show the speedup at different number of environment steps, while the second table shows the throughput capacity. Results suggest that when the number of environments is small (<32), NAVIX CPU is more or as efficient as on GPU, and allows for a larger batch size due to the higher memory capacity (RAM) compared to the GPU memory capacity (VRAM). However, as the batch size grows, the GPU scales better. Its performance remains constant up until around $2^12$ environments, after which starts to grow linearly.

This confirms that the performance gains of NAVIX derive from a systemic transition to a different framework in JAX, and not only from the transition to different types of accelerators. The higher throughput of NAVIX on CPU derives from a higher RAM capacity compared to the VRAM of the GPU.

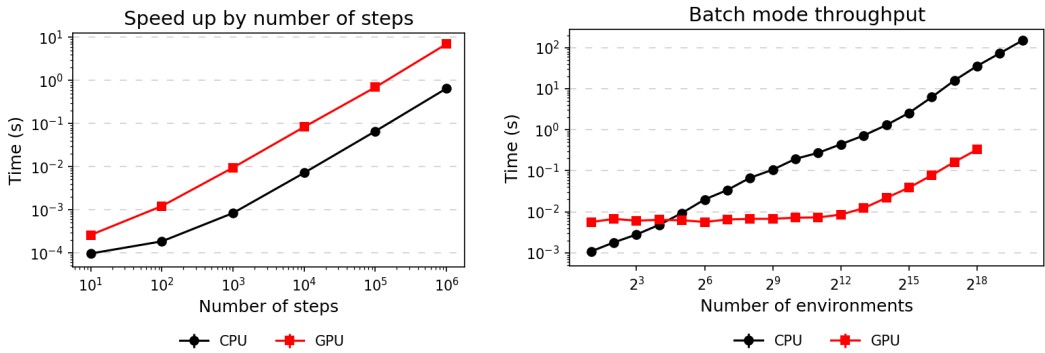

Figure 13: Computation speed across number different number of sequential steps between a CPU- and GPU-compiled NAVIX program.

Figure 14: Computation costs across growing batch sizes between a CPU- and GPU-compiled NAVIX program.

# E    Details on baselines

Here we report the hyperparameters used to run the baselines in Section 4.3, for each algorithm. To allow the table to fit into the page horizontally we shorten the names of each environment and present a full mapping below:

1. FR: `Navix-FourRooms-v0`,
2. DK8: `Navix-DoorKey-8x8-v0`,
3. DK16: `Navix-DoorKey-16x16-v0`,
4. GD6: `Navix-GoToDoor-6x6-v0`,
5. SC9: `Navix-SimpleCrossingS9N1-v0`,
6. E6: `Navix-Empty-6x6-v0`,
7. E16: `Navix-Empty-16x16-v0`,
8. ER8: `Navix-Empty-Random-8x8-v0`,
9. LG6: `Navix-LavaGapS6-v0`,
10. KC43: `Navix-KeyCorridorS4R3-v0`,
11. DO6R: `Navix-Dynamic-Obstacles-6x6-Random-v0`,
12. DS2: `Navix-DistShift2-v0`.

| PPO | DS2 | DK16 | DK8 | DO6R | E16 | E6 | ER8 | FR | GD6 | KC43 | LG6 | SC9 |
|---|---|---|---|---|---|---|---|---|---|---|---|---|
| activation | swish | tanh | tanh | tanh | tanh | relu | swish | swish | tanh | tanh | tanh | swish |
| num_envs | 256 | 16 | 16 | 16 | 16 | 16 | 64 | 128 | 32 | 128 | 16 | 64 |
| num_steps | 64 | 128 | 128 | 128 | 128 | 256 | 128 | 32 | 32 | 64 | 128 | 128 |
| num_epochs | 16 | 2 | 2 | 16 | 2 | 4 | 8 | 2 | 8 | 4 | 8 | 4 |
| num_minibatches | 1 | 1 | 1 | 8 | 1 | 16 | 1 | 8 | 32 | 1 | 8 | 4 |
| learning_rate | 0.0003 | 0.0003 | 0.0003 | 0.0003 | 0.0003 | 0.0003 | 0.0003 | 0.0003 | 0.0003 | 0.0003 | 0.0003 | 0.0003 |
| max_grad_norm | 10 | 1 | 1 | 5 | 1 | 1 | 10 | 5 | 10 | 10 | 0.5 | 0.5 |
| total_timesteps | 1M | 1M | 1M | 1M | 1M | 1M | 1M | 1M | 1M | 1M | 1M | 1M |
| eval_freq | 2K | 2K | 2K | 2K | 2K | 2K | 2K | 2K | 2K | 2K | 2K | 2K |
| gamma | 0.95 | 0.95 | 0.95 | 0.99 | 0.95 | 0.99 | 0.99 | 0.99 | 0.99 | 0.95 | 0.99 | 0.99 |
| gae_lambda | 0.95 | 0.9 | 0.9 | 0.99 | 0.9 | 0.9 | 0.8 | 0.99 | 0.95 | 0.95 | 0.99 | 0.9 |
| clip_eps | 0.2 | 0.2 | 0.2 | 0.2 | 0.2 | 0.2 | 0.2 | 0.2 | 0.2 | 0.2 | 0.2 | 0.2 |
| ent_coef | 0.01 | 0.01 | 0.01 | 0.01 | 0.01 | 0.01 | 0.01 | 0.01 | 0.01 | 0.01 | 0.01 | 0.01 |
| vf_coef | 0.5 | 0.5 | 0.5 | 0.5 | 0.5 | 0.5 | 0.5 | 0.5 | 0.5 | 0.5 | 0.5 | 0.5 |
| normalize_observations | true | false | false | true | false | true | true | false | true | true | true | true |

Table 7: Hyperparameters for the PPO agent. Numeric abbreviations: 1M = 1 048 576, 256K = 262 144.

| SAC | DS2 | DK16 | DK8 | DO6R | E16 | E6 | ER8 | FR | GD6 | KC43 | LG6 | SC9 |
|---|---|---|---|---|---|---|---|---|---|---|---|---|
| activation | swish | swish | swish | tanh | relu | tanh | tanh | tanh | tanh | relu | swish | tanh |
| num_envs | 128 | 128 | 128 | 128 | 128 | 128 | 128 | 128 | 128 | 128 | 128 | 128 |
| buffer_size | 128K | 128K | 128K | 128K | 128K | 128K | 128K | 128K | 128K | 128K | 128K | 128K |
| fill_buffer | 8K | 8K | 8K | 8K | 8K | 8K | 8K | 8K | 8K | 8K | 8K | 8K |
| batch_size | 256 | 128 | 128 | 512 | 512 | 512 | 512 | 256 | 256 | 128 | 1028 | 1028 |
| learning_rate | 0.0003 | 0.0003 | 0.0003 | 0.0003 | 0.0003 | 0.0003 | 0.0003 | 0.0003 | 0.0003 | 0.0003 | 0.0003 | 0.0003 |
| num_epochs | 128 | 128 | 128 | 128 | 128 | 128 | 128 | 128 | 128 | 128 | 128 | 128 |
| total_timesteps | 1M | 1M | 1M | 1M | 1M | 1M | 1M | 1M | 1M | 1M | 1M | 1M |
| eval_freq | 128K | 128K | 128K | 128K | 128K | 128K | 128K | 128K | 128K | 128K | 128K | 128K |
| gamma | 0.8 | 0.8 | 0.8 | 0.8 | 0.8 | 0.8 | 0.95 | 0.9 | 0.9 | 0.8 | 0.8 | 0.8 |
| polyak | 0.9 | 0.995 | 0.995 | 0.95 | 0.995 | 0.95 | 0.9 | 0.995 | 0.995 | 0.95 | 0.9 | 0.9 |
| target_entropy_ratio | 0.6 | 0.9 | 0.9 | 0.5 | 0.5 | 0.5 | 0.3 | 0.9 | 0.9 | 0.4 | 0.6 | 0.6 |
| normalize_observations | true | false | false | false | false | false | false | false | false | true | true | true |

Table 8: Hyperparameters for the SAC agent. Numeric abbreviations: 1M = 1 048 576, 128K = 131 072, 8K = 8 192.

| DQN | DS2 | DK16 | DK8 | DO6R | E16 | E6 | ER8 | FR | GD6 | KC43 | LG6 | SC9 |
|---|---|---|---|---|---|---|---|---|---|---|---|---|
| activation | swish | relu | relu | relu | swish | swish | swish | relu | swish | swish | swish | tanh |
| num_envs | 10 | 10 | 10 | 10 | 10 | 10 | 10 | 10 | 10 | 10 | 10 | 10 |
| num_epochs | 1 | 1 | 1 | 1 | 1 | 1 | 1 | 1 | 1 | 1 | 1 | 1 |
| buffer_size | 128K | 128K | 64K | 128K | 128K | 128K | 128K | 128K | 128K | 128K | 128K | 128K |
| fill_buffer | 8K | 8K | 8K | 8K | 8K | 8K | 8K | 8K | 8K | 8K | 8K | 8K |
| batch_size | 1,024 | 512 | 128 | 2,048 | 1,024 | 1,024 | 1,024 | 256 | 1,024 | 128 | 512 | 512 |
| learning_rate | 0.0003 | 0.0003 | 0.0003 | 0.0003 | 0.0003 | 0.0003 | 0.0003 | 0.0003 | 0.0003 | 0.0003 | 0.0003 | 0.0003 |
| max_grad_norm | 0.5 | 1 | 10 | 0.5 | 10 | 0.5 | 1 | 10 | 0.5 | 10 | 1 | 1 |
| total_timesteps | 1M | 1M | 1M | 1M | 1M | 1M | 1M | 1M | 1M | 1M | 1M | 1M |
| eval_freq | 128K | 128K | 16K | 128K | 128K | 128K | 128K | 128K | 128K | 128K | 128K | 128K |
| gamma | 0.95 | 0.95 | 0.95 | 0.95 | 0.95 | 0.95 | 0.9 | 0.9 | 0.95 | 0.995 | 0.9 | 0.95 |
| eps_start | 1.0 | 1.0 | 1.0 | 1.0 | 1.0 | 1.0 | 1.0 | 1.0 | 1.0 | 1.0 | 1.0 | 1.0 |
| eps_end | 0.01 | 0.01 | 0.05 | 0.01 | 0.01 | 0.01 | 0.05 | 0.01 | 0.01 | 0.01 | 0.05 | 0.05 |
| exploration_fraction | 0.3 | 0.5 | 0.5 | 0.5 | 0.5 | 0.3 | 0.1 | 0.1 | 0.3 | 0.3 | 0.1 | 0.3 |
| target_update_freq | 8K | 2K | 2K | 512 | 1K | 8K | 1K | 2K | 8K | 8K | 4K | 512 |
| ddqn | true | true | true | true | true | true | true | true | true | true | true | true |
| normalize_observations | true | false | false | true | true | true | false | false | true | false | true | false |

Table 9: Hyperparameters for the DQN agent. Numeric abbreviations: 1M = 1 048 576, 128K = 131 072, 64K = 65 536, 16K = 16 384, 8K = 8 192, 4K = 4 096, 2K = 2 048.

| IQN | DS2 | DK16 | DK8 | DO6R | E16 | E6 | ER8 | FR | GD6 | KC43 | LG6 | SC9 |
|---|---|---|---|---|---|---|---|---|---|---|---|---|
| activation | relu | tanh | tanh | tanh | swish | relu | swish | relu | relu | relu | swish | tanh |
| num_envs | 10 | 10 | 10 | 10 | 10 | 10 | 10 | 10 | 10 | 10 | 10 | 10 |
| num_epochs | 1 | 1 | 1 | 1 | 1 | 1 | 1 | 1 | 1 | 1 | 1 | 1 |
| buffer_size | 128K | 128K | 128K | 128K | 128K | 128K | 128K | 128K | 128K | 128K | 128K | 128K |
| fill_buffer | 8K | 8K | 8K | 8K | 8K | 8K | 8K | 8K | 8K | 8K | 8K | 8K |
| batch_size | 512 | 256 | 256 | 256 | 256 | 512 | 256 | 512 | 256 | 256 | 512 | 128 |
| learning_rate | 0.0003 | 0.0003 | 0.0003 | 0.0003 | 0.0003 | 0.0003 | 0.0003 | 0.0003 | 0.0003 | 0.0003 | 0.0003 | 0.0003 |
| kappa | 1.0 | 0.5 | 0.5 | 0.5 | 3.0 | 1.0 | 3.0 | 2.0 | 0.5 | 0.5 | 1.0 | 2.0 |
| num_tau_samples | 32 | 16 | 16 | 16 | 16 | 32 | 16 | 32 | 16 | 16 | 16 | 32 |
| num_tau_prime_samples | 64 | 64 | 64 | 64 | 64 | 64 | 64 | 16 | 64 | 64 | 16 | 64 |
| total_timesteps | 1M | 1M | 1M | 1M | 1M | 1M | 1M | 1M | 1M | 1M | 1M | 1M |
| eval_freq | 128K | 128K | 128K | 128K | 128K | 128K | 128K | 128K | 128K | 128K | 128K | 128K |
| gamma | 0.9 | 0.9 | 0.9 | 0.9 | 0.9 | 0.9 | 0.9 | 0.95 | 0.9 | 0.9 | 0.9 | 0.9 |
| eps_start | 1.0 | 1.0 | 1.0 | 1.0 | 1.0 | 1.0 | 1.0 | 1.0 | 1.0 | 1.0 | 1.0 | 1.0 |
| eps_end | 0.01 | 0.01 | 0.01 | 0.01 | 0.1 | 0.1 | 0.1 | 0.05 | 0.01 | 0.01 | 0.1 | 0.05 |
| exploration_fraction | 0.1 | 0.1 | 0.1 | 0.1 | 0.3 | 0.1 | 0.3 | 0.7 | 0.1 | 0.1 | 0.7 | 0.1 |
| target_update_freq | 1K | 512 | 512 | 512 | 4K | 1K | 4K | 512 | 512 | 512 | 2K | 1K |
| normalize_observations | false | true | true | true | true | false | true | false | true | true | false | true |

Table 10: Hyperparameters for the IQN agent. Numeric abbreviations: 1M = 1 048 576, 128K = 131 072, 8K = 8 192, 4K = 4 096, 2K = 2 048, 1K = 1 024.

| PQN | DS2 | DK16 | DK8 | DO6R | E16 | E6 | ER8 | FR | GD6 | KC43 | LG6 | SC9 |
|---|---|---|---|---|---|---|---|---|---|---|---|---|
| num_envs | 16 | 512 | 512 | 512 | 512 | 16 | 128 | 32 | 128 | 128 | 8 | 128 |
| num_steps | 128 | 128 | 128 | 128 | 16 | 128 | 16 | 512 | 16 | 512 | 512 | 16 |
| num_epochs | 8 | 16 | 16 | 16 | 1 | 8 | 2 | 1 | 2 | 1 | 16 | 2 |
| num_minibatches | 128 | 64 | 64 | 64 | 128 | 128 | 128 | 128 | 128 | 256 | 64 | 128 |
| learning_rate | 0.000246 | 0.000126 | 0.000126 | 0.000126 | 0.000536 | 0.000246 | 0.000104 | 0.000179 | 0.000179 | 0.000211 | 0.000242 | 0.000104 |
| max_grad_norm | 5 | 1 | 1 | 1 | 0.5 | 5 | 1 | 10 | 1 | 1 | 5 | 1 |
| total_timesteps | 1M | 1M | 1M | 1M | 1M | 1M | 1M | 1M | 1M | 1M | 1M | 1M |
| eval_freq | 256K | 256K | 256K | 256K | 256K | 256K | 256K | 256K | 256K | 256K | 256K | 256K |
| gamma | 0.9 | 0.8 | 0.8 | 0.8 | 0.8 | 0.9 | 0.99 | 0.99 | 0.99 | 0.95 | 0.8 | 0.99 |
| td_lambda | 0.8 | 0.2 | 0.2 | 0.2 | 0.8 | 0.8 | 0.8 | 0.4 | 0.8 | 0.8 | 0.8 | 0.8 |
| eps_start | 1 | 1 | 1 | 1 | 1 | 1 | 1 | 1 | 1 | 1 | 1 | 1 |
| eps_end | 0.1 | 0.1 | 0.1 | 0.1 | 0.1 | 0.1 | 0.05 | 0.05 | 0.05 | 0.05 | 0.1 | 0.05 |
| exploration_fraction | 0.3 | 0.3 | 0.3 | 0.3 | 0.5 | 0.3 | 0.2 | 0.7 | 0.2 | 0.4 | 0.2 | 0.2 |
| normalize_observations | false | true | true | true | false | false | true | true | true | true | true | true |

Table 11: Hyperparameters for the PQN agent. Columns use the short env IDs defined in App. E. Numeric abbreviations: 1M = 1 048 576, 256K = 262 144.

| Algorithm | Fitted hyperparameters |
|---|---|
| PPO | #envs, #steps, #epochs, #minibatches, discount factor, $\lambda$ (GAE), grad. norm clip, norm. obs., activation function |
| DQN | batch size, target network update freq., discount factor, exploration fraction, final $\epsilon$, grad. norm clip, norm. obs., activation function |
| SAC | batch size, discount factor, $\tau$ (Polyak update), target entropy ratio, norm. obs., activation function |

Table 12: Fitted hyperparameters for PPO, DQN, and SAC. Details on each hyperparameter set, for each environment and each algorithm are available at `https://github.com/epignatelli/navix/tree/speedup/baselines/rejax/configs`.

# F  Reusable patterns

Here we provide some useful patterns that users can reuse as-they-are or modify to suit their needs. In particular, we show how to jit the full interaction loop of a NAVIX environment in Code 2, and how to run multiple seeds in parallel in Code 3. Further examples, including how to jit a whole training loop with a JAX-based agent, and how to automate hyperparameter search, are available in the NAVIX documentation at `https://epignatelli/navix/examples/getting_started.html`.

## F.1  Jitting full interaction loops

```python
import navix as nx

# init a NAVIX environment
env = nx.make("Navix-KeyCorridorS6R3-v0")

# sample a starting state
timestep = env.reset(key)

# jitting the step function
step_env = jax.jit(env.step)

# unroll the environment for 1000 steps
timestep, _ = jax.lax.scan(
    lambda timestep, _: (unroll(timestep, i % 6), ()),
    timestep,
    (timestep, jnp.arange(1000))
)
```

Code 2: Example code to jit a `Navix-Empty-5x5-v0` environment.

## F.2  Running multiple seeds in parallel

```python
import navix as nx

env = nx.make("Navix-KeyCorridorS6R3-v0")

# define the run function
def run(key):
    def step(state, action):
        timestep, key = state
        key, subkey = jax.random.split(key)
        action = jax.random.randint(subkey, (), 0, env.action_space.n)
        return (env.step(timestep, action), key), ()

    # unroll the environment for 1000 steps
    timestep = env.reset(key)
    timestep, _ = jax.lax.scan(
        step,
        timestep,
        ((timestep, key) jnp.arange(1000)),
    )
    return timestep

seeds = jax.random.split(jax.random.PRNGKey(0), 1000)
batched_end_steps = jax.jit(jax.vmap(run))(seeds)
```

Code 3: Example code to jit a `Navix-Empty-5x5-v0` environment.

## G   Customising NAVIX environments

NAVIX is designed to be highly customisable, allowing users to create new environments by combining existing entities and systems. In this section, we provide examples of how to customise NAVIX environments by using different *systems*.

For example, to create a new environment where the agent has to reach a goal while avoiding lava, we can combine the `Goal` and `Lava` entities with the `Reward` system:

```python
import navix a nx

reward_fn = nx.rewards.compose(
    nx.rewards.on_goal_reached(),
    nx.rewards.on_lava_fall()
)

env = nx.make(
    "Navix-Empty-5x5-v0",
    reward_fn=reward_fn)
```

Code 4: Example code to create a `Navix-Empty-5x5-v0` environment with a custom reward function. See Table 5 for a list of implemented reward functions.

Alternatively, to use a different observation function, we can use the `Observation` system:

```python
import navix as nx

env = nx.make(
    "Navix-Empty-5x5-v0",
    observation_fn=nx.observations.rgb())
```

Code 5: Example code to create a `Navix-Empty-5x5-v0` environment with a custom observation function. See Table 4 for a list of implemented observation functions.

Finally, to terminate the environment, for example, only when the agent reaches the goal, but not when it falls into the lava, we can use the `Termination` system:

```python
import navix as nx

env = nx.make(
    "Navix-Empty-5x5-v0",
    termination_fn=nx.terminations.on_goal_reached())
```

Code 6: Example code to create a `Navix-Empty-5x5-v0` environment with a custom termination function. See Table 6 for a list of implemented termination functions.

These examples can be extended to create more complex environments by combining different systems for the same environment configuration.

## H   Extending NAVIX environments

NAVIX is designed to be easily extensible. Users can create new entities, components, systems, and full environments by implementing the necessary functions. In this section, we provide **templates** to extend NAVIX environments. In particular, Code 7 shows how to create a custom environment, Code 8 shows how to create a custom component, Code 9 shows how to create a custom entity, and Code 10 shows how to create custom systems.

```
import jax, navix as nx

class CustomEnv(nx.Environment):
    def _reset(self, key: jax.Array) -> nx.Timestep:
        """Reset the environment."""
        # create your grid, place your entities, define your mission
        return timestep

nx.registry.register_env(
    "CustomEnv",
    lambda *args, **kwargs: CustomEnv.create(
        observation_fn=nx.observations.symbolic(),
        reward_fn=nx.rewards.on_goal_reached(),
        termination_fn=nx.terminations.on_goal_reached(),
    )
)
```

Code 7: Example code to extend NAVIX by creating a custom environment. The `_reset` function allows to generate a custom starting state, after which the environment will evolve according to the usual systems: intervention, transition, reward and termination functions. Notice that it is convenient to use the environment constructor `create` to automatically set non-orthogonal properties (e.g. observation space and observation function).

```
import jax, navix as nx

class CustomComponent(nx.Componnet):
    """My custom component."""

    custom_property: jax.Array = nx.components.field(shape=())
```

Code 8: Example code to extend NAVIX by creating a custom component. Notice that the property must have a type annotation and specify a shape.

```
import jax, navix as nx

class CustomEntity(nx.Entity, CustomComponent):
    """My custom entity."""

    @property
    def walkable(self) -> jax.Array:
        return jnp.broadcast_to(jnp.asarray(False), self.shape)

    @property
    def transparent(self) -> jax.Array:
        return jnp.broadcast_to(jnp.asarray(False), self.shape)

    @property
    def sprite(self) -> jax.Array:
        sprite = # the address of your sprite, e.g., SPRITES_REGISTRY[Entities.WALL]
        return jnp.broadcast_to(sprite[None], (*self.shape, *sprite.shape))

    @property
    def tag(self) -> jax.Array:
        entity_id = # the id of your entity, e.g., EntityIds.WALL
        return jnp.broadcast_to(entity_id, self.shape)
```

Code 9: Example code to extend NAVIX by creating a custom entity. Notice that four properties must be implemented: `walkable`, `transparent`, `sprite`, and `tag`.

```python
import jax, navix as nx

def my_reward_function(state: nx.State, action: nx.Action, new_state: nx.State) -> jax.Array:
    """My custom reward function."""
    # do stuff
    return reward   # f32[]

def my_termination_function(state: nx.State, action: nx.Action, new_state: nx.State) -> jax.Array:
    """My custom termination function."""
    # do stuff
    return termination   # bool[]

def my_observation_function(state: nx.State) -> jax.Array:
    """My custom observation function."""
    # do stuff
    return observation   # f32[]

def my_intervention_function(state: nx.State, action: nx.Action) -> nx.State:
    """My custom intervention function."""
    # do stuff
    return new_state   # State

def my_transition_function(state: nx.State) -> nx.State:
    """My custom transition function."""
    # do stuff
    return new_state   # State
```

Code 10: Example code to extend NAVIX by creating custom systems.

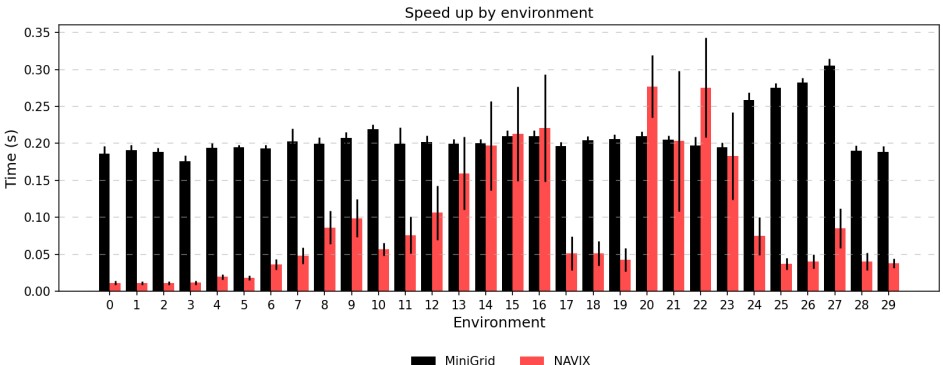

Figure 15: **Ablation.** Speedup of NAVIX compared to the original Minigrid implementation without batching. The identifiers on the x-axis correspond to the environments as reported in Table 13. Lower is better.

# I Additional Tables

| X tick | Env id |
|--------|--------|
| 0 | Navix-Empty-5x5-v0 |
| 1 | Navix-Empty-6x6-v0 |
| 2 | Navix-Empty-8x8-v0 |
| 3 | Navix-Empty-16x16-v0 |
| 4 | Navix-Empty-Random-5x5 |
| 5 | Navix-Empty-Random-6x6 |
| 6 | Navix-DoorKey-5x5-v0 |
| 7 | Navix-DoorKey-6x6-v0 |
| 8 | Navix-DoorKey-8x8-v0 |
| 9 | Navix-DoorKey-16x16-v0 |
| 10 | Navix-FourRooms-v0 |
| 11 | Navix-KeyCorridorS3R1-v0 |
| 12 | Navix-KeyCorridorS3R2-v0 |
| 13 | Navix-KeyCorridorS3R3-v0 |
| 14 | Navix-KeyCorridorS4R3-v0 |
| 15 | Navix-KeyCorridorS5R3-v0 |
| 16 | Navix-KeyCorridorS6R3-v0 |
| 17 | Navix-LavaGapS5-v0 |
| 18 | Navix-LavaGapS6-v0 |
| 19 | Navix-LavaGapS7-v0 |
| 20 | Navix-SimpleCrossingS9N1-v0 |
| 21 | Navix-SimpleCrossingS9N2-v0 |
| 22 | Navix-SimpleCrossingS9N3-v0 |
| 23 | Navix-SimpleCrossingS11N5-v0 |
| 24 | Navix-Dynamic-Obstacles-5x5 |
| 25 | Navix-Dynamic-Obstacles-6x6 |
| 26 | Navix-Dynamic-Obstacles-8x8 |
| 27 | Navix-Dynamic-Obstacles-16x16 |
| 28 | Navix-DistShift1-v0 |
| 29 | Navix-DistShift2-v0 |

Table 13: Correspondence between the x-ticks in Figure 3 and the environment ids.

Table of environments available in NAVIX.

| Env-id | Class | Height | Width | Reward |
|--------|-------|--------|-------|--------|
| Navix-Empty-5x5-v0 | Empty | 5 | 5 | $R_1$ |
| Navix-Empty-6x6-v0 | Empty | 6 | 5 | $R_1$ |
| Navix-Empty-8x8-v0 | Empty | 8 | 8 | $R_1$ |
| Navix-Empty-16x16-v0 | Empty | 16 | 16 | $R_1$ |
| Navix-Empty-Random-5x5 | Empty | 5 | 5 | $R_1$ |
| Navix-Empty-Random-6x6 | Empty | 6 | 6 | $R_1$ |
| Navix-Empty-Random-8x8 | Empty | 8 | 8 | $R_1$ |
| Navix-Empty-Random-16x16 | Empty | 16 | 16 | $R_1$ |
| Navix-DoorKey-5x5-v0 | DoorKey | 5 | 5 | $R_1$ |
| Navix-DoorKey-6x6-v0 | DoorKey | 6 | 6 | $R_1$ |
| Navix-DoorKey-8x8-v0 | DoorKey | 8 | 8 | $R_1$ |
| Navix-DoorKey-16x16-v0 | DoorKey | 16 | 16 | $R_1$ |
| Navix-DoorKey-Random-5x5 | DoorKey | 5 | 5 | $R_1$ |
| Navix-DoorKey-Random-6x6 | DoorKey | 6 | 6 | $R_1$ |
| Navix-DoorKey-Random-8x8 | DoorKey | 8 | 8 | $R_1$ |
| Navix-DoorKey-Random-16x16 | DoorKey | 16 | 16 | $R_1$ |
| Navix-FourRooms-v0 | FourRooms | 17 | 17 | $R_1$ |
| Navix-KeyCorridorS3R1-v0 | KeyCorridor | 3 | 7 | $R_1$ |
| Navix-KeyCorridorS3R2-v0 | KeyCorridor | 5 | 7 | $R_1$ |
| Navix-KeyCorridorS3R3-v0 | KeyCorridor | 7 | 7 | $R_1$ |
| Navix-KeyCorridorS4R3-v0 | KeyCorridor | 10 | 10 | $R_1$ |
| Navix-KeyCorridorS5R3-v0 | KeyCorridor | 13 | 13 | $R_1$ |
| Navix-KeyCorridorS6R3-v0 | KeyCorridor | 16 | 16 | $R_1$ |
| Navix-LavaGap-S5-v0 | LavaGap | 5 | 5 | $R_2$ |
| Navix-LavaGap-S6-v0 | LavaGap | 6 | 6 | $R_2$ |
| Navix-LavaGap-S7-v0 | LavaGap | 7 | 7 | $R_2$ |
| Navix-Crossings-S9N1-v0 | Crossings | 9 | 9 | $R_2$ |
| Navix-Crossings-S9N2-v0 | Crossings | 9 | 9 | $R_2$ |
| Navix-Crossings-S9N3-v0 | Crossings | 9 | 9 | $R_2$ |
| Navix-Crossings-S11N5-v0 | Crossings | 11 | 11 | $R_2$ |
| Navix-Dynamic-Obstacles-5x5 | Dynamic-Obstacles | 5 | 5 | $R_3$ |
| Navix-Dynamic-Obstacles-5x5 | Dynamic-Obstacles | 5 | 5 | $R_3$ |
| Navix-Dynamic-Obstacles-6x6 | Dynamic-Obstacles | 6 | 6 | $R_3$ |
| Navix-Dynamic-Obstacles-6x6 | Dynamic-Obstacles | 6 | 6 | $R_3$ |
| Navix-Dynamic-Obstacles-8x8 | Dynamic-Obstacles | 8 | 8 | $R_3$ |
| Navix-Dynamic-Obstacles-16x16 | Dynamic-Obstacles | 16 | 16 | $R_3$ |
| Navix-DistShift1-v0 | DistShift | 6 | 6 | $R_2$ |
| Navix-DistShift2-v0 | DistShift | 8 | 8 | $R_2$ |
| Navix-GoToDoor-5x5-v0 | GoToDoor | 5 | 5 | $R_1$ |
| Navix-GoToDoor-6x6-v0 | GoToDoor | 6 | 6 | $R_1$ |
| Navix-GoToDoor-8x8-v0 | GoToDoor | 8 | 8 | $R_1$ |

Table 14: List of environments available in NAVIX. *Env-id* denotes the id to instantiate the environment. Here, $R_1$ is the reward function for goal achievement – 1 when the agent is on the green square, and 0 otherwise. $R_2$ is the reward function for goal achievement and lava avoidance – 1 when the agent is on the green square, −1 when the agent is on the lava square, and 0 otherwise. $R_3$ is the reward function for goal achievement and dynamic obstacles avoidance – 1 when the agent is on the green square, −1 when the agent is hit by a flying object, and 0 otherwise. All environments terminate when the reward is not 0, for example, on goal achievement, or on lava collision.

