# OpenReview forum: "NAVIX: Scaling MiniGrid Environments with JAX"
_NeurIPS.cc/2025/Datasets_and_Benchmarks_Track — NeurIPS 2025 Datasets and Benchmarks Track poster_

### Official Review · Reviewer_W2HA · 2025-06-23

**Rating:** 5
**Confidence:** 3

**Summary:**

Solving large-scale worlds by RL is a good topic and could be of interest to many researchers.
In existing RL experiments, interactions are typically computed on the CPU, limiting training speed and throughput.
To address this problem, this paper re-implements MiniGrid in JAX with GPU. i.e., Nvidia A100 80 GB.
The new environment achieves over 160000× speed improvements in batch mode, supporting up to 2048 agents in parallel on a single
 Nvidia A100 80 GB.

**Dataset Code Accessibility:**

Yes

**Dataset Code Comments:**

The author has open-source the project on GitHub.
There is sufficient detail on data collection and organization, availability and maintenance, and ethical and responsible use. The authors also provide the source code to support reproducibility.

**Ethical Considerations:**

No, there are no or only very minor ethics concerns

**Final Justification:**

My main concerns have been addressed. I keep my positive score.

**Limitations Weaknesses:**

1. The main concern is the lack of the experiments. The performance of RL algorithms for POMDP tasks should be given, e.g., the performance of Dreamer.
2. It would be beneficial to provide the parameter setting of the testing RL methods.

**Strengths Contributions:**

1. The paper is well structured and clearly written.
2. Overall, I like this paper, the idea in this paper is quite intuitive, solving large-scale worlds is a good topic and could be of interest to many researchers.
3. The new environment can effectively test the existing RL method with high throughput, e.g., high dimensional and multi-agent systems. NAVIX can be over 160000 faster than the original Minigrid implementation, turning 1-week
 experiments into 15-minute ones.
4. The new environment includes MDP tasks and POMDP tasks.

---

> ### Author Rebuttal · Authors · 2025-07-31
>
> We would like to thank reviewer W2HA for the feedback on the paper and for suggesting acceptance with a score of 5.
>
> The review positively highlights that the paper is “well structured and clearly written” and of “interest to many researcher”, and understand the concerns raised, which we aim to address below.
>
> —
>
> > *The main concern is the lack of the experiments. The performance of RL algorithms for POMDP tasks should be given, e.g., the performance of Dreamer.*
>
> Thank you for highlighting this. The main purpose of our study is to present an environment that replicates the semantics of MiniGrid, but orders of magnitude faster. We provided evidence of the claim in four experiments, whose results are shown in Figure 3, 4, 5, and 6. To reinforce the argument, we have now included additional experiments:
> - Testing the performance of NAVIX on CPU compared to GPU, to marginalise over the new design pattern choice (the ECS model), and shed light on whether the performance gains derive from the transition to a highly parallelised accelerator, or to a new design paradigm using JAX functional API. Results are shown in Table 1.
> - Testing the performance on consumer-grade hardware – a desktop with an i7-11700 @ 2.50GHz CPU, 32Gb of RAM, and an NVidia RTX A4000 with 16Gb of VRAM – to give readers a more comprehensive expectation of performance on more commonly available hardware. Results are shown in Table 2.
> - Testing the impact of the Markovianity of the reward function. In Table 3 we report the average task completion rate at convergence and standard deviation across 32 seeds. Results shows that, for some environments, in particular those where PPO converges to near-optimality, the success rate often presents higher variance with non-Markov rewards.
>
> —
>
> > *It would be beneficial to provide the parameter setting of the testing RL methods.*
>
> Thank you for spotting this. Configuration parameters for each algorithm in the baselines can be found in the original NAVIX repository (they are quite verbose and would exceed the character limits of this rebuttal) under the `baselines/rejax/configs`. All experiments have been carried on the `speedup` branch. Unfortunately, we are not allowed to use links in the rebuttal, and as a markdown table, it would take roughly 3200 character per baseline, exceeding the character allowance for this rebuttal, which is why we are not including them in this response. We would be keen to include them also in an additional table of the supplementary material.
>
> —
>
> We would like to thank again the reviewer for the useful feedback on the paper. We have added 3 new experiments to address the concern about the lack of experimental evidence, and clarified details on the parameter settings for the baselines. Should the reviewer consider the main concerns addressed, we would be grateful if they would increase their score to reflect the progress.

---

> > ### Comment · Reviewer_W2HA · 2025-08-01
> >
> > Thank the authors for their rebuttal, which addresses my concerns. I keep my positive score.

---

> > > ### Author Response · Authors · 2025-08-01
> > > **Performance of additional RL algorithms**
> > >
> > > Dear reviewer W2HA,
> > >
> > > Thank you for keeping your positive score. In addition to the experiments mentioned above, we have added the performance of new RL algorithms, namely Implicit Quantile Network (IQN) and Parallel Q-Network (PQN). In the following two tables we show the return at termination for $\approx$ 1M steps of training averaged across 5 seeds. We are not allowed to update our codebase during the rebuttal, as per NeurIPS guidelines, but we will add the hyperparameters used for each algorithm and for each environment as soon as a decision is made.
> > > ### PQN
> > >
> > > | Env/Step                      | 0        | 131k      | 262k      | 393k      | 524k      | 655k      | 786k      | 918k      | 1.05M     |
> > > |:-----------------------------|:---------|:---------:|:---------:|:---------:|:---------:|:---------:|:---------:|:---------:|:---------:|
> > > | SimpleCrossingS9N1-v0        | 0.00±0.00 | 0.00±0.00 | 0.01±0.02 | 0.14±0.28 | 0.18±0.34 | 0.18±0.34 | 0.18±0.34 | 0.18±0.34 | 0.18±0.34 |
> > > | DistShift2-v0                | 0.20±0.40 | 1.00±0.00 | 1.00±0.00 | 1.00±0.00 | 1.00±0.00 | 1.00±0.00 | 1.00±0.00 | 1.00±0.00 | 1.00±0.00 |
> > > | DoorKey-16x16-v0             | 0.00±0.00 | 0.00±0.00 | 0.00±0.00 | 0.00±0.00 | 0.00±0.00 | 0.00±0.00 | 0.00±0.00 | 0.00±0.00 | 0.00±0.00 |
> > > | DoorKey-8x8-v0               | 0.00±0.00 | 0.00±0.00 | 0.00±0.00 | 0.00±0.00 | 0.00±0.00 | 0.00±0.00 | 0.00±0.00 | 0.00±0.00 | 0.00±0.00 |
> > > | Dynamic-Obstacles-6x6-Random-v0 | 0.01±0.02 | 0.21±0.04 | 0.38±0.12 | 0.35±0.11 | 0.36±0.11 | 0.36±0.11 | 0.36±0.11 | 0.36±0.11 | 0.36±0.11 |
> > > | Empty-16x16-v0               | 0.00±0.00 | 0.00±0.00 | 0.00±0.00 | 0.00±0.00 | 0.00±0.00 | 0.00±0.00 | 0.00±0.00 | 0.00±0.00 | 0.00±0.00 |
> > > | Empty-6x6-v0                 | 0.00±0.00 | 1.00±0.00 | 1.00±0.00 | 0.83±0.34 | 1.00±0.00 | 1.00±0.00 | 1.00±0.00 | 1.00±0.00 | 1.00±0.00 |
> > > | Empty-Random-8x8-v0          | 0.01±0.02 | 0.05±0.03 | 0.08±0.04 | 0.19±0.06 | 0.21±0.08 | 0.21±0.08 | 0.21±0.08 | 0.21±0.08 | 0.21±0.08 |
> > > | FourRooms-v0                 | 0.00±0.00 | 0.00±0.00 | 0.00±0.00 | 0.00±0.00 | 0.00±0.00 | 0.00±0.00 | 0.00±0.00 | 0.00±0.00 | 0.00±0.00 |
> > > | GoToDoor-6x6-v0              | 0.03±0.04 | 0.22±0.13 | 0.26±0.16 | 0.21±0.12 | 0.34±0.13 | 0.34±0.13 | 0.34±0.13 | 0.34±0.13 | 0.34±0.13 |
> > > | KeyCorridorS4R3-v0           | 0.00±0.00 | 0.00±0.00 | 0.00±0.00 | 0.00±0.00 | 0.00±0.00 | 0.00±0.00 | 0.00±0.00 | 0.00±0.00 | 0.00±0.00 |
> > > | LavaGapS6-v0                 | 0.00±0.00 | 0.23±0.22 | 0.45±0.12 | 0.74±0.16 | 0.53±0.17 | 0.53±0.17 | 0.53±0.17 | 0.53±0.17 | 0.53±0.17 |
> > >
> > > ### IQN
> > >
> > > | Env/Step                      | 0        | 131k      | 262k      | 393k      | 524k      | 655k      | 786k      | 918k      | 1.05M     |
> > > |:-----------------------------|:---------|:---------:|:---------:|:---------:|:---------:|:---------:|:---------:|:---------:|:---------:|
> > > | SimpleCrossingS9N1-v0        | 0.00±0.00 | 0.82±0.09 | 0.80±0.07 | 0.90±0.09 | 0.92±0.14 | 1.00±0.00 | 0.99±0.02 | 1.00±0.00 | 1.00±0.00 |
> > > | DistShift2-v0                | 0.19±0.38 | 1.00±0.00 | 1.00±0.00 | 1.00±0.00 | 1.00±0.00 | 1.00±0.00 | 1.00±0.00 | 1.00±0.00 | 1.00±0.00 |
> > > | DoorKey-16x16-v0             | 0.00±0.00 | 0.00±0.00 | 0.00±0.00 | 0.00±0.00 | 0.00±0.00 | 0.00±0.00 | 0.00±0.00 | 0.00±0.00 | 0.00±0.00 |
> > > | DoorKey-8x8-v0               | 0.00±0.00 | 0.00±0.00 | 0.00±0.00 | 0.00±0.00 | 0.00±0.00 | 0.00±0.00 | 0.00±0.00 | 0.00±0.00 | 0.00±0.00 |
> > > | Dynamic-Obstacles-6x6-Random-v0 | 0.05±0.08 | 0.70±0.11 | 0.64±0.11 | 0.50±0.03 | 0.62±0.14 | 0.54±0.09 | 0.58±0.15 | 0.61±0.09 | 0.64±0.10 |
> > > | Empty-16x16-v0               | 0.00±0.00 | 0.00±0.00 | 0.00±0.00 | 0.20±0.40 | 0.19±0.38 | 0.20±0.40 | 0.20±0.40 | 0.20±0.40 | 0.20±0.40 |
> > > | Empty-6x6-v0                 | 0.02±0.04 | 1.00±0.00 | 1.00±0.00 | 1.00±0.00 | 1.00±0.00 | 1.00±0.00 | 1.00±0.00 | 1.00±0.00 | 1.00±0.00 |
> > > | Empty-Random-8x8-v0          | 0.02±0.04 | 0.30±0.11 | 0.83±0.15 | 0.97±0.04 | 0.99±0.02 | 1.00±0.00 | 0.99±0.02 | 1.00±0.00 | 1.00±0.00 |
> > > | FourRooms-v0                 | 0.01±0.02 | 0.01±0.02 | 0.05±0.03 | 0.07±0.07 | 0.01±0.02 | 0.00±0.00 | 0.02±0.02 | 0.01±0.02 | 0.02±0.02 |
> > > | GoToDoor-6x6-v0              | 0.08±0.09 | 0.88±0.13 | 0.75±0.15 | 0.84±0.09 | 0.79±0.09 | 0.83±0.15 | 0.83±0.08 | 0.78±0.08 | 0.82±0.10 |
> > > | KeyCorridorS4R3-v0           | 0.00±0.00 | 0.00±0.00 | 0.00±0.00 | 0.00±0.00 | 0.00±0.00 | 0.00±0.00 | 0.00±0.00 | 0.00±0.00 | 0.00±0.00 |
> > > | LavaGapS6-v0                 | 0.00±0.00 | 0.02±0.04 | 0.97±0.06 | 1.00±0.00 | 1.00±0.00 | 1.00±0.00 | 1.00±0.00 | 0.99±0.02 | 1.00±0.00 |

---

### Official Review · Reviewer_GQ6s · 2025-06-25

**Rating:** 4
**Confidence:** 2

**Summary:**

This paper introduces NAVIX, a MiniGrid environment toolkit rebuilt with JAX. By moving computation to GPU/TPU accelerators and adopting a modular design called Entity-Component-System Model (ECSM), it achieves efficient reinforcement learning execution. The paper explains NAVIX's design patterns and principles. Experiments show NAVIX is 160,000 times faster than original MiniGrid, reducing experiments from weeks to 15 minutes. It also keeps full compatibility with MiniGrid's interface and uses two reward designs to match the original environment's behavior.

**Dataset Code Accessibility:**

Yes

**Ethical Considerations:**

No, there are no or only very minor ethics concerns

**Final Justification:**

I keep my original rating.

**Limitations Weaknesses:**

1. The paper offers Markovian/non-Markovian reward options but doesn't test if they produce consistent policy behaviors. This risks replication failures when reproducing prior MiniGrid studies, weakening NAVIX's value as a drop-in replacement.
2. Critical experimental details, such as the specific environments used and configuration parameters, are not fully described.
3. The claimed 160,000x speedup is not benchmarked against other JAX-based frameworks. Since JAX implementations vary significantly in optimization strategies, comparing only with the original CPU-based MiniGrid cannot prove whether NAVIX’s speedup stems from JAX’s general advantages or its unique design.
4. While NAVIX shows massive hardware throughput gains, no analysis confirms this reduces end-to-end training time.

**Strengths Contributions:**

1. Refactoring MiniGrid using JAX and an ECS (Entity-Component-System) model is a novel and promising solution.
2. NAVIX achieves 44× faster execution and over 1,000,000× higher throughput versus CPU-based MiniGrid by leveraging JIT compilation and GPU acceleration. This is a major benefit for DRL experimentation.
3. Supports multiple accelerators (GPUs/TPUs), simplifying deployment compared to traditional distributed systems.
4. One GPU runs 2,048 parallel environments, eliminating network delays and reducing costs vs. multi-machine setups.

---

> ### Author Rebuttal · Authors · 2025-07-31
>
> We would like to thank reviewer GQ6s for the helpful feedback on the manuscript, and for suggesting acceptance with a score of 4.
>
> We are glad that the review finds that NAVIX is a “*novel and promising solution*”, providing “*44× faster execution and over 1,000,000× higher throughput [...] is a major benefit for DRL experimentation.*”, "*eliminating network delays and reducing costs vs. multi-machine setups*”, and “*simplifying deployment compared to traditional distributed systems.*”
>
> We also acknowledge the concerns raised and we have taken the following actions to address them.
>
> —
>
> > *The paper offers Markovian/non-Markovian reward options but doesn't test if they produce consistent policy behaviors. This risks replication failures when reproducing prior MiniGrid studies, weakening NAVIX's value as a drop-in replacement.*
>
> We agree that particular care should be taken by researchers aiming to reproduce MiniGrid experiments on NAVIX. To shed lights on the impact of reward Markovianity on the performance of RL training, we have added a new experiment that compares the performance of PPO trained with the two reward functions. In Table R1 below, we report the task completion rate at convergence across 32 seeds. Results show that, for some environments, in particular those where PPO converges to near-optimality, the success rate often presents higher variance with non-Markov rewards.
> | Table R1 | Navix-Empty-6x6-v0 | Navix-Empty-Random-6x6-v0 | Navix-Empty-16x16-v0 | Navix-Dynamic-Obstacles-6x6-Random-v0 | Navix-DoorKey-5x5-v0 | Navix-DoorKey-16x16-v0 | Navix-DistShift2-v0 | Navix-GoToDoor-6x6-v0 | Navix-GoToDoor-8x8-v0 | Navix-FourRooms-v0 | Navix-LavaGapS6-v0 | Navix-LavaGapS7-v0 |
> | --- | --- | --- | --- | --- | --- | --- | --- | --- | --- | --- | --- | --- |
> | Markov | 1.000 ± 0.001 | 0.867 ± 0.052 | 0.000 ± 0.000 | 0.412 ± 0.042 | 0.762 ± 0.149 | 0.000 ± 0.000 | 0.969 ± 0.174 | 0.851 ± 0.051 | 0.000 ± 0.000 | 0.000 ± 0.000 | 0.999 ± 0.001 | 0.404 ± 0.488 |
> | Non-Markov | 0.531 ± 0.499 | 0.405 ± 0.313 | 0.000 ± 0.000 | 0.414 ± 0.052 | 0.609 ± 0.126 | 0.000 ± 0.000 | 0.500 ± 0.500 | 0.048 ± 0.044 | 0.000 ± 0.000 | 0.000 ± 0.000 | 0.687 ± 0.463 | 0.000 ± 0.000 |
>
> —
>
> > *Critical experimental details, such as the specific environments used and configuration parameters, are not fully described.*
>
> Thank you for spotting this. For Figure 4, 5 and 6, we used MiniGrid-Empty-8x8-v0 with default parameters: symbolic observations; Markovian reward function that yields +1 when the goal is reached and 0 otherwise; termination when the reward is +1.
>
> Configuration parameters for each algorithm in the baselines can be found in the original NAVIX repository (they are quite verbose and would exceed the character limits of this rebuttal) under the `baselines/rejax/configs`. All experiments have been carried on the `speedup` branch. Unfortunately, we are not allowed to use links in the rebuttal, and as a markdown table, it would take roughly 3200 character per baseline, exceeding the character allowance for this rebuttal, which is why we are not including them in this reponse. We would be keen to include them also in an additional table of the supplementary material.
>
> —
>
> > *The claimed 160,000x speedup is not benchmarked against other JAX-based frameworks. Since JAX implementations vary significantly in optimization strategies, comparing only with the original CPU-based MiniGrid cannot prove whether NAVIX’s speedup stems from JAX’s general advantages or its unique design.*
>
> The speedup of NAVIX does not only depend on JAX, but also on the transition to an ECS design pattern. It is hard to disentangle the impact of each on the performance. Even if other JAX-based environment reimplementations might not use an ECS model, they are still required to migrate to a different design pattern due to the transition from a stateful to a stateless paradigm. This makes it hard to separate the effects of JAX from those of the new design patterns. To marginalise the performance gains over the design patter, we have added a new experiment comparing the performance of NAVIX on CPU and GPU: the exact same Python program, with the exact same design pattern, but compiled and run on different accelerators. The experiment is carried on a desktop with an i7-11700 @ 2.50GHz CPU, 32Gb of RAM, and an Nvidia RTX A4000 with 16Gb of VRAM. Table 2 shows the speedup at different number of environment steps, while the second table shows the throughput capacity. Results show that when the number of environments is small (<32), NAVIX CPU is more or as efficient as on GPU. This confirms that the performance gains of NAVIX derive from a systemic transition to a different framework in JAX, and not only from the transition to different types of accelerators. The higher throughput of NAVIX on CPU derives from a higher RAM capacity compared to the VRAM of the GPU.
>
> | Table R2 | 10 | 100 | 1000 | 10000 | 100000 | 1000000 |
> | --- | --- | --- | --- | --- | --- | --- |
> | GPU | 0.000 ± 0.000 | 0.001 ± 0.000 | 0.010 ± 0.000 | 0.083 ± 0.003 | 0.696 ± 0.001 | 7.025 ± 0.015 |
> | CPU | 0.000 ± 0.000 | 0.000 ± 0.000 | 0.001 ± 0.000 | 0.007 ± 0.000 | 0.067 ± 0.000 | 0.654 ± 0.002 |
> | Speedup | 0.4 ± 0.1× | 0.2 ± 0.0× | 0.1 ± 0.0× | 0.1 ± 0.0× | 0.1 ± 0.0× | 0.1 ± 0.0× |
>
>
> | Table 3 | 2 | 4 | 8 | 16 | 32 | 64 | 128 | 256 | 512 | 1024 | 2048 | 4096 | 8192 | 16384 | 32768 | 65536 | 131072 | 262144 | 524288 | 1048576 |
> | --- | --- | --- | --- | --- | --- | --- | --- | --- | --- | --- | --- | --- | --- | --- | --- | --- | --- | --- | --- | --- |
> | GPU | 0.006 ± 0.001 | 0.007 ± 0.001 | 0.006 ± 0.001 | 0.006 ± 0.001 | 0.006 ± 0.001 | 0.006 ± 0.001 | 0.006 ± 0.001 | 0.007 ± 0.001 | 0.007 ± 0.001 | 0.007 ± 0.001 | 0.007 ± 0.001 | 0.009 ± 0.001 | 0.012 ± 0.001 | 0.022 ± 0.001 | 0.039 ± 0.001 | 0.077 ± 0.006 | 0.163 ± 0.008 | 0.328 ± 0.011 | - | - |
> | CPU | 0.001 ± 0.000 | 0.002 ± 0.000 | 0.003 ± 0.000 | 0.005 ± 0.000 | 0.009 ± 0.001 | 0.020 ± 0.000 | 0.034 ± 0.001 | 0.067 ± 0.000 | 0.106 ± 0.001 | 0.194 ± 0.000 | 0.273 ± 0.001 | 0.441 ± 0.001 | 0.718 ± 0.001 | 1.292 ± 0.029 | 2.563 ± 0.012 | 6.191 ± 0.471 | 15.843 ± 0.500 | 35.668 ± 0.374 | 72.910 ± 0.216 | 149.680 ± 0.100 |
>
> —
>
> > *While NAVIX shows massive hardware throughput gains, no analysis confirms this reduces end-to-end training time.*
>
> We report the typical training times of a single and multiple PPO agents in parellel in Figure 6. These are training time, including the costs of environment steps, experience evaluation, and policy and value function updates. The horizontal, black, dashed line in Figure 6 represents the cost of training a single PPO agent on MiniGrid, which sits around 240s for 1M environment steps. The red line shows the same cost for NAVIX, whose times even at 2048 PPO agents in parallel (49s) are still lower than those of MiniGrid for one single agent.
>
> —
>
> Once again, we would like to thank the reviewer for the valuable feedback to improve our paper. We have taken action to address the concerns raised by:
> - Adding a new experiment that investigates the impact of Markovianity on the PPO performance.
> - Adding a new experiment that compares the performance of NAVIX on GPU and CPU, to marginalise the performance gain over the transition to a new design pattern, compared to MiniGrid.
> - Clarified the configuration parameters, and the missing details for each experiment
> - Clarified that Figure 6 shows the end-to-end performance gain of training a PPO agent on NAVIX.
>
> Should the reviewer consider the concerns addressed, we would be grateful it if they would increase their score to reflect the improvements.

---

> > ### Comment · Reviewer_GQ6s · 2025-08-05
> >
> > My concerns have been addressed well.

---

> > > ### Author Response · Authors · 2025-08-06
> > >
> > > Dear reviewer GQ6s,
> > >
> > > Thank you for the follow-up comment. We are glad to hear that the concerns have been well addressed, and we appreciate your time and feedback to improve our manuscript.

---

### Official Review · Reviewer_eXye · 2025-07-01

**Rating:** 4
**Confidence:** 4

**Summary:**

This paper introduces NAVIX, a JAX-based reimplementation of MiniGrid to support large-scale parallelism. Compared to previous work, NAVIX faithfully reproduces the original MiniGrid, with the addition of an optional purely Markovian reward function. Benchmarks show that NAVIX achieves orders-of-magnitude speedups over CPU-only MiniGrid. This work also provides baseline results, including PPO, DQN, and SAC.

**Additional Feedback:**

1. Baseline benchmark: This work only provides baseline results without further insights. As NAVIX exactly replicates MiniGrid, these benchmark results can also be obtained—albeit slowly—using the original MiniGrid environments. This weakens the direct contribution of NAVIX unless its acceleration is better leveraged. I recommend including experiments like those that specifically explore the effects of increased parallelism on training performance across different algorithms. Such analysis would showcase NAVIX’s strengths and make a stronger case for its utility.
2. Stateful vs. stateless: If I understand correctly, a function must be stateless to be jittable (see [official document](https://docs.jax.dev/en/latest/stateful-computations.html#a-general-strategy)). Thus, certain wording may be incorrect in the paper. For example:
   - In Line 171, "For environments to be fully jittable, the computation must be stateful." should be "... must be stateless".
   - and in Line 100, "Among others, the obstacles to transform a stateless *(should be stateful)* program, where a function is allowed to change elements that are not an input of the function, to a stateful *(should be stateless)* one"
3. Formatting issues:
   - Avoid hyperlinking technical terms like "Deep RL" or "MDP" to empty links.
   - Expressions like $4.4 \times 10^1$ are unnecessarily verbose and should be simplified to 44.
   - "Minigrid−like" should be "Minigrid-like"

**Dataset Code Accessibility:**

Partly

**Dataset Code Comments:**

The code for the environments is clearly available. However, it appears that the baseline implementations (especially DDQN and SAC) are not well-documented and open-sourced, as I have been unable to find relevant code in the public GitHub repository.

**Ethical Comments:**

The work does not involve human subjects or sensitive data.

**Ethical Considerations:**

No, there are no or only very minor ethics concerns

**Final Justification:**

The position of this work has been well clarified, thus the major concern has been addressed.

**Limitations Weaknesses:**

The main concern lies in the potential impact and broader contribution to the community. Reinforcement learning remains sample-inefficient. I believe instead of relying only on scaling environment parallelism, we need to develop more sample-efficient learning algorithms with real-world applicability and efficiency. This work may offer limited scientific insight without additional contributions beyond the fact that in such symbolic environments like MiniGrid, we still need $10^5$ environment steps and can only rely on parallel environments to fastly learn it.

I am clearly aware that different reviewers have different subjective opinions on the importance of MiniGrid for the RL community. I will refer to other reviews and authors' rebuttals to make my final rating.

**Strengths Contributions:**

This work represents a solid engineering effort in faithfully replicating MiniGrid in JAX with such a significant acceleration.

---

> ### Author Rebuttal · Authors · 2025-07-31
>
> We thank reviewer eXye for the useful feedback on the manuscript.
>
> We are glad to hear that the review recognises that NAVIX “*faithfully replicating MiniGrid in JAX with such a significant acceleration.*”, which is the main claim of our manuscript, as stated in Section 1.
>
> We also acknowledge the concerns raised in the review, and we have taken actions to address them.
>
>
> —
>
> > *The main concern lies in the potential impact and broader contribution to the community. Reinforcement learning remains sample-inefficient. I believe instead of relying only on scaling environment parallelism, we need to develop more sample-efficient learning algorithms with real-world applicability and efficiency.
> This work may offer limited scientific insight without additional contributions beyond the fact that in such symbolic environments like MiniGrid, we still need environment steps and can only rely on parallel environments to fastly learn it.
> I am clearly aware that different reviewers have different subjective opinions on the importance of MiniGrid for the RL community. I will refer to other reviews and authors' rebuttals to make my final rating.*
>
> and
>
> > *Baseline benchmark: This work only provides baseline results without further insights. As NAVIX exactly replicates MiniGrid, these benchmark results can also be obtained—albeit slowly—using the original MiniGrid environments. This weakens the direct contribution of NAVIX unless its acceleration is better leveraged. I recommend including experiments like those that specifically explore the effects of increased parallelism on training performance across different algorithms. Such analysis would showcase NAVIX’s strengths and make a stronger case for its utility.*
>
>
>
> While we agree with the reviewer that RL remains sample-inefficient, we would like to clarify that we are not presenting NAVIX as a workaround for sample-inefficiency, but rather as a mechanism to amplify experimentation and research. Research conducted in NAVIX shares the same ultimate goals as research conducted in MiniGrid (and other environments), as expressed by the reviewer: "*to develop more sample-efficient learning algorithms with real-world applicability and efficiency*.
>
> There have been a number of recent works that have conducted their research on similar JAX-based vectorized environments, which uncovered research insights that benefit RL in a general sense: [1, 2, 3]. The insights uncovered by these works were because of the speed of the JAX-based implementations used.
>
> Furthermore, RL environments alone have been the main contribution of many publications both at NeurIPS [4, 5], especially in the Datasets and Benchmarks track, and at other top tier venues [6, 7, 8, 9].
>
> Given the points above, and the relevance of MiniGrid in RL experimentation, as also identified by reviewer VjGM (“MiniGrid is the foundation for a vast amount of research [...]”), we respectfully disagree that NAVIX offers "limited scientific insights" – most importantly since NAVIX is successful at "faithfully replicating MiniGrid in JAX", as stated by the reviewer themselves.
>
> To summarize: we agree more research is needed to make RL more sample-efficient, but we argue that NAVIX is complementary to this effort, and in fact is a very fast and useful tool for advancing RL research along that dimension.
>
> [1] Lu, C., Kuba, J., Letcher, A., Metz, L., Schroeder de Witt, C., & Foerster, J. (2022). Discovered policy optimisation. Advances in Neural Information Processing Systems, 35, 16455-16468.
>
> [2] Dizdarević, T., Hammond, R., Gessler, T., Calinescu, A., Cook, J., Gallici, M., ... & Foerster, J. N. (2025). Ad-Hoc Human-AI Coordination Challenge. arXiv preprint arXiv:2506.21490.
>
> [3] Mayor, W., Obando-Ceron, J., Courville, A., & Castro, P. S. (2025). The Impact of On-Policy Parallelized Data Collection on Deep Reinforcement Learning Networks. arXiv preprint arXiv:2506.03404.
>
> [4] Küttler, H., Nardelli, N., Miller, A., Raileanu, R., Selvatici, M., Grefenstette, E., & Rocktäschel, T. (2020). The nethack learning environment. Advances in Neural Information Processing Systems, 33, 7671-7684.
>
> [5] Szot, A., Clegg, A., Undersander, E., Wijmans, E., Zhao, Y., Turner, J., ... & Batra, D. (2021). Habitat 2.0: Training home assistants to rearrange their habitat. Advances in neural information processing systems, 34, 251-266.
>
> [6] Leibo, J. Z., Dueñez-Guzman, E. A., Vezhnevets, A., Agapiou, J. P., Sunehag, P., Koster, R., ... & Graepel, T. (2021, July). Scalable evaluation of multi-agent reinforcement learning with melting pot. In International conference on machine learning (pp. 6187-6199). PMLR.
>
> [7] Bard, N., Foerster, J. N., Chandar, S., Burch, N., Lanctot, M., Song, H. F., ... & Bowling, M. (2020). The hanabi challenge: A new frontier for AI research. Artificial Intelligence, 280, 103216.
>
> [8] Cobbe, Karl, Chris Hesse, Jacob Hilton, and John Schulman. "Leveraging procedural generation to benchmark reinforcement learning." In International conference on machine learning, pp. 2048-2056. PMLR, 2020.
>
> [9] Maxime Chevalier-Boisvert, Dzmitry Bahdanau, Salem Lahlou, Lucas Willems, Chitwan Saharia, Thien Huu Nguyen, & Yoshua Bengio (2019). BabyAI: First Steps Towards Grounded Language Learning With a Human In the Loop. In International Conference on Learning Representations.
>
> —
>
> > *The code for the environments is clearly available. However, it appears that the baseline implementations (especially DDQN and SAC) are not well-documented and open-sourced, as I have been unable to find relevant code in the public GitHub repository.*
>
>
> We use the Rejax baselines as stated in Section 4.3, and in the NeurIPS Paper Checklist (Section 4). Rejax is an open-source, broadly adopted JAX-based set of RL agents. All experiments have been carried on the `speedup` branch. Instructions on how to reproduce the baselines are in the NAVIX repository. Unfortunately, we are not allowed to use links in this rebuttal comment, so we cannot directly point to a specific url.
>
> —
>
> > *Stateful vs. stateless: If I understand correctly, a function must be stateless to be jittable (see official document). Thus, certain wording may be incorrect in the paper. For example:
> In Line 171, "For environments to be fully jittable, the computation must be stateful." should be "... must be stateless".
> and in Line 100, "Among others, the obstacles to transform a stateless (should be stateful) program, where a function is allowed to change elements that are not an input of the function, to a stateful (should be stateless) one"*
>
> Thank you very much for spotting the mistake. We are keen to revise the manuscript to read the correct wording.
>
> —
>
> > Formatting issues:
> Avoid hyperlinking technical terms like "Deep RL" or "MDP" to empty links.
> Expressions like are unnecessarily verbose and should be simplified to 44.
> "Minigrid−like" should be "Minigrid-like"
>
>
> We are keen to remove hyperlinks to empty links on acronyms, to simplify the unnecessarily verbose expression at lines 50 and 212, and replace the dash in MiniGrid-like with a hyphen, as spotted.
>
> —
>
> We would like to thank once again the reviewer for the valuable feedback to improve our manuscript. We have provided arguments on the major concerns raised in the review, and keen to take action on the edits requested. Given that the review recognises that main claims of the papers are well supported (“*faithfully replicating MiniGrid in JAX with such a significant acceleration.*”), provided the reviewer is satisfied with our responses., we would appreciate it if they would increase their score to reflect this.

---

> > ### Comment · Reviewer_eXye · 2025-08-03
> >
> > I am well convinced by the rebuttal addressing my major concern, as well as the comments from Reviewer VjGM—particularly the point that faster environment simulation helps rapid experimentation and iteration in DRL research.
> >
> > As a result, I have decided to increase my score.

---

> > > ### Author Response · Authors · 2025-08-06
> > >
> > > Dear reviewer eXye,
> > >
> > > Thank you for the follow-up and for revisiting your evaluation. We are glad to hear that the rebuttal well addresses your concerns, and we would like to thank you for increasing your score, suggesting acceptance.
> > >
> > > We understand updated ratings should not be visible to authors according to the NeurIPS guidelines (quote below), but we can still see "Score: 3", suggesting that the score update did not take effect yet.
> > > To avoid technical obstacles, could we kindly ask reviewer eXye whether the new score is already in place? Thank you in advance.
> > >
> > > > *Please note that when the reviewer updates the “Rating”, you will not be able to see it (the rating will disappear for you) until the final decisions are out. This mechanism is implemented on purpose to ensure reviewers are free of unduly pressure to modify the score. Instead, try to discuss your rebuttal with the reviewer who may indicate if they are satisfied with it.*

---

### Official Review · Reviewer_VjGM · 2025-07-02

**Rating:** 4
**Confidence:** 2

**Summary:**

This paper introduces NAVIX, a JAX-based reimplementation of the MiniGrid environment suite, to overcome the slow, CPU-bound limitations of the original version. By leveraging hardware accelerators like GPUs, NAVIX achieves massive speedups of over 160,000x in batch mode, drastically reducing experiment times from a week to just 15 minutes.

**Dataset Code Accessibility:**

Yes

**Dataset Code Comments:**

Code can be found on github.

**Ethical Considerations:**

No, there are no or only very minor ethics concerns

**Final Justification:**

The additional data provided by the author in the rebuttal has resolved my concerns. I will maintain my positive rating.

**Limitations Weaknesses:**

The paper repeatedly emphasizes that NAVIX is a "drop-in replacement" for MiniGrid. However, due to JAX's pure-function and stateless nature, NAVIX's API, which requires explicitly passing timestep and key, is fundamentally different from Gymnasium's API.

The astounding 160,000x speedup was achieved on a top-tier Nvidia A100 80GB GPU. To give readers a more comprehensive expectation of performance, providing a set of comparison data on a more mainstream consumer-grade GPU (e.g., an RTX 4070) would greatly enhance the universality and persuasiveness of the results.

There is a minor calculation error on page 7: The paper states $1 \times 1M / 240s = 4.2 \times 10^4$, but the correct result should have an exponent of 3, not 4.

**Strengths Contributions:**

The paper clearly identifies that inefficient environment simulation is a major bottleneck restricting rapid experimentation and iteration in DRL research. NAVIX directly solves this problem, making it a very practical contribution to the community. MiniGrid is the foundation for a vast amount of research in areas like exploration, curriculum learning, and representation learning, meaning that improving MiniGrid's performance will accelerate this wide range of research fields.

The performance improvement claimed by the paper is not incremental but an order-of-magnitude leap. The ability to "reduce experiment times from one week to 15 minutes" can fundamentally change researchers' workflows, making larger-scale and more complex experiments feasible.

Using JAX is the most suitable technical choice for this goal. The paper fully leverages JAX's core advantages, such as the XLA optimization from just-in-time (JIT) compilation and the efficient vectorization achieved through vmap.

The paper provides benchmark results for mainstream algorithms like PPO, DDQN, and SAC across multiple NAVIX environments, which serves as sufficient experimental validation and a strong argument for its effectiveness.

---

> ### Author Rebuttal · Authors · 2025-07-31
>
> We thank reviewer VjGM for the constructive and insightful feedback on the manuscript, and for suggesting acceptance with a score of 4.
>
> The review highlights that “*NAVIX directly solves this problem, making it a very practical contribution to the community*”, and that NAVIX “*can fundamentally change researchers' workflows, making larger-scale and more complex experiments feasible.*” with an “*astounding 160,000x speedup*”.
> We appreciate that the reviewer finds that the “benchmark results for mainstream algorithms like PPO, DDQN, and SAC across multiple NAVIX environments serve as sufficient experimental validation and a strong argument for its effectiveness.”
>
> We also agree with the three major limitations that the review identified and that addressing them will greatly improve the manuscript. To address them we have taken the following three actions.
> > *The paper repeatedly emphasizes that NAVIX is a ‘drop-in replacement’ for MiniGrid. However, due to JAX's pure-function and stateless nature, NAVIX's API, which requires explicitly passing timestep and key, is fundamentally different from Gymnasium's API.*
>
> We agree that our phrasing was imprecise. NAVIX is not a drop‑in replacement at the API level. Instead, NAVIX reimplements the MiniGrid task suite with identical semantics (observations, actions, rewards, and terminations) but exposes a JAX-native, functional interface. This design – passing state, timestep, and RNG keys explicitly – allows NAVIX to JIT‑compile the entire agent–environment loop, which is the primary source of our large throughput gains. We are keen to revise the manuscript to:
> remove the “drop‑in” wording
> state clearly that NAVIX is semantically compatible with MiniGrid but uses a different API by design
>
>
> > *The astounding 160,000x speedup was achieved on a top-tier Nvidia A100 80GB GPU. To give readers a more comprehensive expectation of performance, providing a set of comparison data on a more mainstream consumer-grade GPU (e.g., an RTX 4070) would greatly enhance the universality and persuasiveness of the results.*
>
> We have added new experiments testing the speedup, and throughput improvements of NAVIX on a consumer-grade machine. The experiment is carried on a desktop with an i7-11700 @ 2.50GHz CPU, 32Gb of RAM, and an NVidia RTX A4000 with 16Gb of VRAM (we did not have an available RTX 4070, unfortunately, but the A4000 should be slightly slower, albeit with 4GB more memory). All results are averaged across $5$ runs, and shown in the following tables (Table R1, R2, R3, R4). Following the same calculations of the paper at line 243-248, the speedup of NAVIX on a consumer-grade machine is: $\frac{512 \text{agents} \times 16 \text{environments} \times 1M \text{steps}}{58.18s} \frac{1M \text{steps}}{240s}$. Here, $512$ is the number of PPO agents, $16$ the number of environments for each agent, $1M$ the number of training steps. We conservatively used the MiniGrid computation times for the high-end machine ($240s$), rather than the consumer-grade one. In short, NAVIX yields a speedup of $\simeq 34,000\times$ on a consumer-grade machine, confirming better speed and better higher throughput.
> | Table R1 | Navix-Empty-5x5-v0 | Navix-Empty-6x6-v0 | Navix-Empty-8x8-v0 | Navix-Empty-16x16-v0 | Navix-Empty-Random-5x5-v0 | Navix-Empty-Random-6x6-v0 | Navix-DoorKey-5x5-v0 | Navix-DoorKey-6x6-v0 |
> | --- | --- | --- | --- | --- | --- | --- | --- | --- |
> | NAVIX | 0.010 ± 0.001 | 0.010 ± 0.001 | 0.010 ± 0.001 | 0.010 ± 0.001 | 0.017 ± 0.002 | 0.016 ± 0.002 | 0.031 ± 0.006 | 0.036 ± 0.008 |
> | MiniGrid | 0.130 ± 0.007 | 0.127 ± 0.007 | 0.125 ± 0.003 | 0.122 ± 0.004 | 0.131 ± 0.004 | 0.128 ± 0.006 | 0.128 ± 0.004 | 0.131 ± 0.007 |
> | Speedup | 13.1 ± 1.5× | 12.6 ± 1.5× | 12.7 ± 1.4× | 12.2 ± 1.2× | 7.9 ± 1.1× | 8.0 ± 1.0× | 4.1 ± 0.8× | 3.6 ± 0.8× |
>
> | Table R2 | 10 | 100 | 1K | 10K | 100K | 1M |
> | --- | --- | --- | --- | --- | --- | --- |
> | NAVIX | 0.00 ± 0.00 | 0.00 ± 0.00 | 0.01 ± 0.00 | 0.09 ± 0.00 | 0.72 ± 0.04 | 7.06 ± 0.06 |
> | MiniGrid | 0.00 ± 0.00 | 0.01 ± 0.00 | 0.12 ± 0.00 | 1.23 ± 0.01 | 12.30 ± 0.06 | 123.61 ± 0.17 |
> | Speedup | 2.8 ± 2.6× | 7.2 ± 4.4× | 12.1 ± 1.3× | 14.3 ± 0.5× | 17.1 ± 1.0× | 17.5 ± 0.1× |
>
> | Table R3 | 2 | 4 | 8 | 16 | 32 | 64 | 128 | 256 | 512 | 1024 | 2048 | 4096 | 8192 | 16384 | 32768 | 65536 | 131072 | 262144 |
> | --- | --- | --- | --- | --- | --- | --- | --- | --- | --- | --- | --- | --- | --- | --- | --- | --- | --- | --- |
> | NAVIX | 0.006 ± 0.001 | 0.006 ± 0.001 | 0.006 ± 0.001 | 0.006 ± 0.001 | 0.006 ± 0.001 | 0.006 ± 0.001 | 0.007 ± 0.001 | 0.007 ± 0.001 | 0.007 ± 0.001 | 0.007 ± 0.001 | 0.007 ± 0.001 | 0.008 ± 0.001 | 0.012 ± 0.001 | 0.021 ± 0.001 | 0.037 ± 0.001 | 0.077 ± 0.003 | 0.163 ± 0.007 | 0.329 ± 0.011 |
> | MiniGrid | 0.252 ± 0.063 | 0.254 ± 0.005 | 0.404 ± 0.020 | 0.605 ± 0.001 | - | - | - | - | - | - | - | - | - | - | - | - | - | - |
>
> | Table R4 | 1 | 2 | 4 | 8 | 16 | 32 | 64 | 128 | 256 | 512 |
> | --- | --- | --- | --- | --- | --- | --- | --- | --- | --- | --- |
> | NAVIX | 12.176 ± 0.657 | 12.311 ± 0.537 | 12.523 ± 0.419 | 12.587 ± 0.449 | 13.108 ± 0.486 | 14.479 ± 0.481 | 17.581 ± 0.505 | 21.862 ± 0.597 | 32.153 ± 0.527 | 58.180 ± 0.479 |
> > *There is a minor calculation error on page 7: The paper states , but the correct result should have an exponent of 3, not 4.*
>
> Thank you very much for spotting the mistake. We are keen to revise the manuscript to read the correct exponent.
>
> Again, we would like to thank the reviewer for the valuable feedback to improve our manuscript. We have taken action to address the concerns raised in the review. We would be grateful it if the reviewer were to increase their score to reflect the changes, should the reviewer consider the concerns addressed.

---

> > ### Comment · Reviewer_VjGM · 2025-08-04
> >
> > Thanks for the reply. I believe my concern has been addressed. Since my original score was already positive, my score will remain unchanged.

---

> > > ### Author Response · Authors · 2025-08-06
> > >
> > > Dear reviewer VjGM,
> > >
> > > We are keen to hear that concerns have been fully addressed and we would like to thank you for keeping a positive score.

---

### Decision · Program_Chairs · 2025-09-18

**Decision:**

Accept (poster)

**Comment:**

This paper introduces NAVIX, a complete reimplementation of the widely used MiniGrid suite of reinforcement learning environments in JAX. The primary scientific claim is that by leveraging JAX for GPU/TPU acceleration, NAVIX can achieve massive simulation speedups, thereby removing a significant bottleneck in RL research. The authors report an over 160,000x speed improvement in batch mode on high-end hardware, which can reduce experiment times from a week to just 15 minutes. This is presented as a crucial infrastructure contribution that enables faster design iterations and more scalable development of RL models.

The paramount strength of this work, unanimously recognized by reviewers, is its immense practical value and potential impact on the RL community. MiniGrid is a foundational benchmark for research in numerous areas, including exploration, curriculum learning, and language-conditioned RL, and the original CPU-bound implementation is a well-known bottleneck. NAVIX directly and effectively solves this problem, offering an orders-of-magnitude performance leap that can fundamentally alter research workflows and enable previously infeasible large-scale experiments. The technical execution is solid, representing a significant and well-executed engineering effort that faithfully replicates MiniGrid's semantics.

Initial reviews raised several valid weaknesses. A core point of debate, highlighted in a confidential note by reviewer VjGM, was whether this work, being a reimplementation, qualified as a scientific contribution or was more of a technical report. Other concerns included the misleading use of the term "drop-in replacement," as the JAX-native API is functionally different from the original Gymnasium API , and the lack of performance benchmarks on more common consumer-grade hardware, which could limit the perceived universality of the results. Reviewers also requested more clarity on reproducing baselines, additional experiments comparing reward function variations, and corrections to minor errors.

The authors provided a comprehensive and effective rebuttal that successfully addressed all major concerns. They immediately conceded that "drop-in replacement" was imprecise and committed to clarifying that NAVIX is semantically compatible but has a different, JAX-native API by design. Crucially, they ran new experiments on a consumer-grade GPU (Nvidia RTX A4000), demonstrating a still-astounding speedup of approximately 34,000x, which fully addressed concerns about performance generalization. They also provided new results comparing Markovian and non-Markovian reward functions, clarified how to reproduce baselines using the Rejax library, and agreed to fix all noted formatting and calculation errors. This thorough response convinced reviewers to maintain or, in one case, increase their scores.